# `Iron`: Private Inference on Transformers

**Meng Hao**[1*] **Hongwei Li**[1†] **Hanxiao Chen**[1] **Pengzhi Xing**[1] **Guowen Xu**[2] **Tianwei Zhang**[2]

[1]University of Electronic Science and Technology of China
[2]Nanyang Technological University
`{menghao,hanxiao.chen,p.xing}@std.uestc.edu.cn`
`hongweili@uestc.edu.cn`
`{guowen.xu,tianwei.zhang}@ntu.edu.sg`

## Abstract

We initiate the study of private inference on Transformer-based models in the client-server setting, where clients have private inputs and servers hold proprietary models. Our main contribution is to provide several new secure protocols for matrix multiplication and complex non-linear functions like Softmax, GELU activations, and LayerNorm, which are critical components of Transformers. Specifically, we first propose a customized homomorphic encryption-based protocol for matrix multiplication that crucially relies on a novel compact packing technique. This design achieves $\sqrt{m}\times$ less communication ($m$ is the number of rows of the output matrix) over the most efficient work. Second, we design efficient protocols for three non-linear functions via integrating advanced underlying protocols and specialized optimizations. Compared to the state-of-the-art protocols, our recipes reduce about half of the communication and computation overhead. Furthermore, all protocols are numerically precise, which preserve the model accuracy of plaintext. These techniques together allow us to implement `Iron`, an efficient Transformer-based private inference framework. Experiments conducted on several real-world datasets and models demonstrate that `Iron` achieves $3 \sim 14\times$ less communication and $3 \sim 11\times$ less runtime compared to the prior art.

## 1 Introduction

Transformer-based models [1–5] have realized tremendous success in the fields of natural language processing (NLP) and computer vision (CV) due to their strong representation capabilities. As a new neural network architecture, the Transformer [1] mainly utilizes the self-attention mechanism to compute representations without sequence-aligned recurrence or convolution. Following this work, a number of Transformer variants, such as BERT [2] and GPT [3] in NLP, ViT [4] and Swin Transformer [5] in CV, have achieved state-of-the-art performance on lots of real-world tasks.

The success of Transformers and other big models facilitates emerging inference services and applications [6, 7]. In particular, a service provider trains a complex model based on the Transformer, and deploys it as a paid inference service, e.g., machine translation and question answering. A client queries this service with his input samples and obtains the desired responses. Unfortunately, current inference systems suffer from serious privacy concerns [8]. On the one hand, clients need to send confidential inputs to the service provider, which could compromise the data privacy of these clients if the provider is untrusted. On the other hand, it is undesirable for the provider to distribute the proprietary Transformer-based model to clients, since he needs a large amount of data and computation resources to construct the model [9]. Therefore, there exists a gap between

---

[*]This work was done at NTU as a visiting student.
[†]Corresponding author

36th Conference on Neural Information Processing Systems (NeurIPS 2022).

unprecedented performance and privacy constraints, which motivates our study of private Transformer inference.

Private inference aims to protect server's model weights from clients, while guaranteeing that the server learns no information about clients' private inputs. Recently, private inference on traditional neural networks (e.g., convolutional neural networks) have been approached by using secure 2-party computation (2PC) techniques [10–14]. However, due to essentially different structures, private Transformer inference brings several new challenges. First, *Transformer-based models use lots of high-dimensional matrix multiplications*, rather than matrix-vector multiplications widely studied in prior works. While we can straightforwardly extend prior matrix-vector multiplication protocols to our setting, unfortunately, even the most efficient design [14] incurs heavy communication because of interacting a large amount of ciphertexts. Second, *Transformer-based models use complex math functions like Softmax, GELU activations [15], and LayerNorm, in each block*, rather than crypto-friendly non-linear functions such as ReLU and Maxpool. Existing methods either use precision-impaired high-order polynomial approximations [16, 11] or only support limited math functions for specific scenarios [17]. Even worse, all of these approaches are computationally intensive and often require a large amount of communication (for more related works, please refer to Appendix A.5). To facilitate the widespread adoption of Transformer-based inference services in privacy-critical scenarios, designing efficient protocols for the above complex operations is of paramount importance.

In this paper, we design `Iron`, an efficient hybrid cryptographic framework for private Transformer inference without revealing any sensitive information about the server's model weights or clients' inputs. `Iron` contributes several new specialized protocols for the complicated operations in Transformers to alleviate the performance overhead. Specifically, we first propose a customized homomorphic encryption-based protocol for matrix multiplications. Our insight is to pack more plaintext inputs into a single ciphertext by devising a compact packing method, while preserving the functionality of matrix multiplication. Compared to the most efficient matrix-vector multiplication solution implemented in Cheetah [14], we can achieve $\sqrt{m}\times$ ($m$ is the number of rows of the output matrix) improvement in terms of communication overhead, which is about $8\times$ reduction for various Transformer models. Second, we carefully design efficient protocols for Softmax, GELU, and LayerNorm. These protocols are built upon SIRNN [17], the state-of-the-art cryptographic framework for private inference on recurrent neural networks, and make several customized optimizations, such as reducing the overhead of exponentiation in Softmax and simplifying GELU and LayerNorm. These optimizations achieve $1.3 \sim 1.8\times$ less runtime and $1.4 \sim 1.8\times$ less communication on three non-linear functions. Furthermore, these protocols are numerically precise, which preserve the model accuracy of plaintext. We also give a formal security proof for our designed protocols to demonstrate the security guarantee.

Based on the above efficient components, we implement a private Transformer inference framework, `Iron`, and conduct end-to-end experiments with various BERT architectures [2] (BERT-Tiny, BERT-Medium, BERT-Base, and BERT-Large) on GLUE benchmarks [18]. Note that `Iron` is readily extended to other Transformer-based models (e.g., ViT) since they share very similar architectures and same operations. Experimental results show that `Iron` achieves $3 \sim 14\times$ less communication and $3 \sim 11\times$ less runtime costs over SIRNN on four BERT models. Moreover, compared with the general-purpose state-of-the-art framework MP-SPDZ [16], `Iron` has up to two orders of magnitude improvement in terms of both communication and computation efficiency.

A concurrent work [19] proposed a privacy-preserving Transformer inference with homomorphic encryption, called THE-X. Below, we illustrate some important differences in terms of protocol design and security. (1) Protocol design. Our work aims to design new efficient protocols for the complex operations of Transformer-based models, while orthogonal to ours, THE-X replaces them with crypto-friendly operations. For example, THE-X replaces GELUs with simpler operations, i.e., ReLUs, and Softmax with the combination of ReLU and polynomials. (2) Security. Our work achieves more rigorous privacy protection than THE-X. Specifically, our work uses homomorphic encryption and secret sharing techniques to hide private information (including intermediate results) of all layers. Such rigorous privacy guarantee is in line with recent state-of-the-art private inference works [14, 17]. However, in THE-X, the inputs of each non-linear layer are leaked to the client, which may causes severe privacy leakages in real-world applications [13]. Therefore, our work may be used to enhance the security of THE-X.

## 2 Preliminaries

### 2.1 Threat Model

As shown in the left part of Figure 1, `Iron` works in a general private inference scenario, where the server $P_0$ holds a Transformer-based model $M$ with private weights $w$, while the client $P_1$ holds a private input $x$. Our framework enables the client to query the server's inference service and learn the output of the model on its input, i.e., $M(w, x)$. Same as prior works [14, 17], we consider an honest-but-curious adversary that passively corrupts either the server or the client, but not both. Such an adversary follows the protocol specification exactly, but may try to learn more information[3] than allowed (e.g., the model's weights or inference inputs) via analyzing the data it receives. In Appendix A.1.2, we give a more formal description of the threat model for security analysis.

### 2.2 Cryptographic Primitives

All our protocols are built on the 2-out-of-2 additive secret sharing (ASS) technique [21, 22] over the ring $\mathbb{Z}_{2^\ell}$, in which an $\ell$-bit input $x$ is split into two random shares $\langle x \rangle_0, \langle x \rangle_1$, held by $P_0$ and $P_1$, respectively, such that $x = \langle x \rangle_0 + \langle x \rangle_1 \mod \mathbb{Z}_{2^\ell}$. When $\ell = 1$, i.e., over $\mathbb{Z}_2$, we use $\langle x \rangle^{\mathsf{B}}$ to denote boolean shares. In our protocols, we use $\langle x \rangle$ to denote that $P_b$ holds $\langle x \rangle_b$ for $b \in \{0, 1\}$. The security [21] guarantees that given a share $\langle x \rangle_0$ or $\langle x \rangle_1$, the value of $x$ is perfectly hidden. This secret-sharing property is maintained throughout our private inference scheme. ASS naturally supports linear operations without communication. For instance, to compute $cx + y$ with the constant $c$ and secret shares $\langle x \rangle$ and $\langle y \rangle$, $P_b$ can locally compute $\langle z \rangle_b = c\langle x \rangle_b + \langle y \rangle_b$, where $\langle z \rangle_0 + \langle z \rangle_1 = cx + y \mod 2^\ell$. To achieve more functionalities under shared inputs, we require to invoke advanced homomorphic encryption or oblivious transfer techniques [22].

**Notations.** Let $x \nmid y$ means $x$ is not a divisor of $y$. We use bold lower-case letters (e.g., $\mathbf{x}$) to represent vectors, and bold upper-case letters (e.g., $\mathbf{X}$) to denote matrices. Like prior works [17, 14], we encode inputs with the fixed-point representation denoted by Fix (refer to Appendix A.1.1 for details). Let $\mathbb{A}_{N,p}$ denote the set of integer polynomials $\mathbb{A}_{N,p} = \mathbb{Z}_p[x] / (x^N + 1)$. We use the circumflex of lower-case letters (e.g., $\hat{a}$) to represent a polynomial, and $\hat{a}[i]$ to denote the $i$-th coefficient of $\hat{a}$. Given polynomials $\hat{x}, \hat{y} \in \mathbb{A}_{N,p}$, the product $\hat{z} = \hat{x} \cdot \hat{y}$ over $\mathbb{A}_{N,p}$ is defined as

$$\hat{z}[i] = \sum_{0 \leq j \leq i} \hat{x}[j]\hat{y}[i - j] - \sum_{i < j < N} \hat{x}[j]\hat{y}[N - j + i] \mod p. \tag{1}$$

**Additively Homomorphic Encryption (AHE)** [23, 24]. This encryption scheme additionally enables linearly homomorphic operations on ciphertexts. Specifically, an AHE scheme is a tuple of algorithms $\mathsf{AHE} = (\mathsf{KeyGen}; \mathsf{Enc}; \mathsf{Dec}; \mathsf{Eval})$ with the parameter $\{N, q, p\}$ and the following syntax: 1) $\mathsf{KeyGen}(1^k) \rightarrow (pk, sk)$: on input a security parameter $\kappa$, $\mathsf{KeyGen}$ is a randomized algorithm that outputs a public key $pk \in \mathbb{A}_{N,q}$ and a secret key $sk \in \mathbb{A}_{N,q}$. 2) $\mathsf{Enc}(pk, \hat{m}) \rightarrow \hat{c}$: the encryption algorithm $\mathsf{Enc}$ takes a plaintext polynomial $\hat{m} \in \mathbb{A}_{N,p}$ and encrypts it using $pk$ into a ciphertext polynomial $\hat{c} \in \mathbb{A}_{N,q}$. 3) $\mathsf{Dec}(sk, \hat{c}) \rightarrow \hat{m}$: on input $sk$ and a ciphertext $\hat{c}$, the (deterministic) decryption algorithm $\mathsf{Dec}$ recovers the plaintext message $\hat{m}$. 4) $\mathsf{Eval}(pk, \hat{c}_1, \hat{c}_2, \mathsf{func}) \rightarrow \hat{c}$: on input $pk$, two ciphertexts $\hat{c}_1, \hat{c}_2$ containing $\hat{m}_1, \hat{m}_2$, and a linear function func, $\mathsf{Eval}$ outputs a new ciphertext $\hat{c}$ encrypting $\mathsf{func}(\hat{m}_1, \hat{m}_2)$. Let $\boxplus, \boxminus, \boxtimes$ denote homomorphic addition, homomorphic subtraction and homomorphic multiplication with a plaintext, respectively. `Iron` builds the matrix multiplication protocol on the Brakerski-Fan-Vercauteren (BFV) scheme [25, 26], which is one of the state-of-the-art lattice-based homomorphic encryption solutions.

**Oblivious Transfer.** The 1-out-of-$k$ oblivious transfer (OT) [27] is denoted by $k\text{-OT}_\ell$, where one party is the sender with $k$ messages $x_0, \ldots, x_{k-1} \in \{0, 1\}^\ell$ and the other party is the receiver with an index $i \in [k]$. The receiver learns $x_i$ as the output, and the sender learns nothing. Additionally, we also use the 1-out-of-2 correlated OT, denoted by $2\text{-COT}_\ell$ [28], which is defined as follows: the sender inputs a correlation $x \in \mathbb{Z}_{2^\ell}$, the receiver inputs a choice bit $i \in \{0, 1\}$, and the protocol outputs a random element $r \in \mathbb{Z}_{2^\ell}$ to the sender and $r + i \cdot x$ to the receiver. $k\text{-OT}_\ell$ and $2\text{-COT}_\ell$ require $2\lambda + k\ell$ and $\lambda + \ell$ bits of communication, respectively, and are executed in 2 rounds. The OT protocols are widely used to build the underlying protocols [13, 17] in Figure 3 we rely on.

---

[3]Like prior works [17, 14], `Iron` does not hide the information that can be indirectly extracted from the inference results. Study of mitigation solutions (e.g., differential privacy [20]) is beyond the scope of this work.

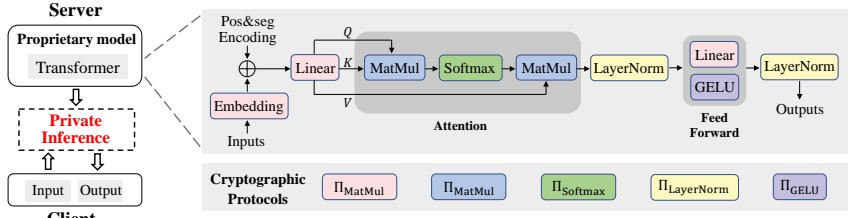

Figure 1: Overview of private Transformer inference

# 3 Overview

## 3.1 Transformer Architecture

The Transformer is an encoder-decoder architecture, where both parts have a similar structure. Hence, we mainly focus on the encoder below. The encoder is composed of a stack of identical blocks, each with two sub-layers, i.e., a multi-head self-attention mechanism and a feed-forward network, as shown in Figure 1. Besides, residual connection and layer normalization (LayerNorm) are employed around each of the two sub-layers. We describe a block of the encoder as follows.

**Attention layer.** An attention function can be described as mapping a query $\mathbf{X}_Q$ and a set of key-value pairs $(\mathbf{X}_K, \mathbf{X}_V)$ to a weighted sum of the values, where the weight is computed by a metric of the query with the corresponding key [1]. This function can be formalized as below:

$$\text{Attention}(\mathbf{X}_Q, \mathbf{X}_K, \mathbf{X}_V) = \text{Softmax}\left(\mathbf{X}_Q \mathbf{X}_K^T / \sqrt{d}\right) \mathbf{X}_V, \tag{2}$$

where $\mathbf{X}_Q, \mathbf{X}_K, \mathbf{X}_V$ are different linear projections of the input $\mathbf{X}$, i.e., $\mathbf{X}_Q = \mathbf{X}\mathbf{W}_Q, \mathbf{X}_K = \mathbf{X}\mathbf{W}_K, \mathbf{X}_V = \mathbf{X}\mathbf{W}_V$, and $d$ is the dimension of representations. Multi-head attention extends the above mechanism to $H$ parallel attention layers and is illustrated in Appendix A.1.3.

**Feed-forward layer.** A fully connected feed-forward layer consists of two linear transformations with a GELU activation in between, where GELU is the Gaussian Error Linear Unit function [15]. This layer can be represented as follows:

$$\text{FeedForward}(\mathbf{X}) = \text{GELU}(\mathbf{X}\mathbf{W}_1 + \mathbf{b}_1)\mathbf{W}_2 + \mathbf{b}_2. \tag{3}$$

In addition to the above encoder-decoder blocks, an embedding layer is employed at the beginning of the model to convert input tokens $\mathbf{X}_{\text{input}}$ to continuous feature vector representations. This is formulated as $\mathbf{X} = \mathbf{X}_{\text{input}}\mathbf{W}_E$, where $\mathbf{W}_E$ is the embedding lookup table.

## 3.2 Private Transformer Inference

According to the required crypographic operations, the layers of Transformers can be broken into two categories - *linear* and *non-linear*.

*1) Linear layers*: these include embedding, matrix multiplication in Attention, and fully-connected layer. All protocols of these operations rely crucially on the matrix multiplication protocol $\Pi_{\text{MatMul}}$ in Section 4.1. It lies in the setting where $P_0$ and $P_1$ take as input the matrices $\mathbf{X}$ and $\mathbf{Y}$, and learn the sharings $\langle\mathbf{Z}\rangle_0$ and $\langle\mathbf{Z}\rangle_1$, respectively, such that $\mathbf{Z} = \mathbf{X}\mathbf{Y}$.

*2) Non-linear layers*: these consist of Softmax, GELU and LayerNorm. The non-linear operations are directly evaluated by exploiting our proposed protocols in Section 4.2, i.e., $\Pi_{\text{GELU}}$, $\Pi_{\text{Softmax}}$, $\Pi_{\text{LayerNorm}}$. All of these take the additive shares $\langle\mathbf{X}\rangle$ as input, and output the additive shares $\langle\mathbf{Y}\rangle = \Pi_{\text{Func}}(\langle\mathbf{X}\rangle)$, where $\text{Func} \in \{\text{GELU}, \text{Softmax}, \text{LayerNorm}\}$.

**Overview of private inference.** In the entire private inference process, we maintain the following invariant: $P_0$ and $P_1$ begin with additive shares of the input to the layer, and end with additive shares (over the same ring $\mathbb{Z}_{2^\ell}$) of the output of the layer after the protocol. This allows us to stitch protocols for arbitrary layers sequentially to obtain a secure computation scheme for any Transformer-based models. To clearly understand our scheme, we take an example of private inference between $P_0$ and $P_1$ on the embedding layer and the first encoder block, as shown in the right part of Figure 1. Here, our example uses the designed protocols as a black box, i.e., $\Pi_{\text{MatMul}}$, $\Pi_{\text{GELU}}$, $\Pi_{\text{Softmax}}$, and $\Pi_{\text{LayerNorm}}$, and we discuss how to efficiently implement these protocols in Section 4.

Figure 2: A toy example of our matrix multiplication protocol.

In the embedding layer, $P_0$ and $P_1$ take as inputs $\mathbf{W}_E$ and $\mathbf{X}_{\text{input}}$, and invoke the matrix multiplication protocol $\Pi_{\text{MatMul}}$ to compute the input embedding $\mathbf{X}$. This is represented as $\langle \mathbf{X} \rangle = \Pi_{\text{MatMul}}(\mathbf{X}_{\text{input}}, \mathbf{W}_E)$. In the attention layer, $\Pi_{\text{MatMul}}$ is also invoked to generate the query, key and value matrices. For instance, $\langle \mathbf{X}_Q \rangle = \Pi_{\text{MatMul}}(\langle \mathbf{X} \rangle_1, \mathbf{W}_Q) + \langle \mathbf{X} \rangle_0 \mathbf{W}_Q$, where $\langle \mathbf{X} \rangle_0 \mathbf{W}_Q$ can be computed locally by $P_0$. The same idea is used in the generation of $\langle \mathbf{X}_K \rangle$ and $\langle \mathbf{X}_V \rangle$. Then, we compute $\mathbf{X}_{QK} = \mathbf{X}_Q \mathbf{X}_K^T$ that requires two invocations of $\Pi_{\text{MatMul}}$. The formulation is $\langle \mathbf{X}_{QK} \rangle = \Pi_{\text{MatMul}}(\langle \mathbf{X}_Q \rangle_0, \langle \mathbf{X}_K \rangle_1) + \Pi_{\text{MatMul}}(\langle \mathbf{X}_Q \rangle_1, \langle \mathbf{X}_K \rangle_0) + \langle \mathbf{X}_Q \rangle_0 \langle \mathbf{X}_K \rangle_0^T + \langle \mathbf{X}_Q \rangle_1 \langle \mathbf{X}_K \rangle_1^T$, where the last two terms can be evaluated by $P_0$ and $P_1$ locally. After that, we evaluate $\langle \widetilde{\mathbf{X}}_{QK} \rangle = \Pi_{\text{Softmax}}(\langle \mathbf{X}_{QK} \rangle / \sqrt{d})$, followed by computing $\langle \widetilde{\mathbf{Z}} \rangle = \langle \widetilde{\mathbf{X}}_{QK} \mathbf{X}_V \rangle$ by two invocations of $\Pi_{\text{MatMul}}$. Then, the two parties invoke the LayerNorm protocol to evaluate $\langle \mathbf{Z} \rangle = \Pi_{\text{LayerNorm}}(\langle \widetilde{\mathbf{Z}} \rangle)$. In the feed-forward layer, they compute $\langle \widetilde{\mathbf{Y}}_1 \rangle = \Pi_{\text{MatMul}}(\langle \mathbf{Z} \rangle_1, \mathbf{W}_1) + \langle \mathbf{Z} \rangle_0 \mathbf{W}_1$, followed by $\langle \widetilde{\mathbf{Y}} \rangle = \Pi_{\text{GELU}}(\langle \widetilde{\mathbf{Y}}_1 \rangle)$. After that, they compute $\langle \widetilde{\mathbf{Y}}_2 \rangle = \Pi_{\text{MatMul}}(\langle \widetilde{\mathbf{Y}} \rangle_1, \mathbf{W}_1) + \langle \widetilde{\mathbf{Y}} \rangle_0 \mathbf{W}_1$, where $\langle \widetilde{\mathbf{Y}} \rangle_0 \mathbf{W}_1$ can be computed locally by $P_0$. Finally, the evaluation of the encoder completes after computing $\langle \mathbf{Y} \rangle = \Pi_{\text{LayerNorm}}(\langle \widetilde{\mathbf{Y}}_2 \rangle)$.

## 4 Supporting Protocols

### 4.1 Protocol for Matrix Multiplication

As shown in Section 1, naively extending well-studied matrix-vector multiplication protocols [29, 13, 14] to our matrix multiplication setting results in a significant communication overhead, mainly due to frequently interacting ciphertexts. In this work, we build a specialized matrix multiplication protocol on top of the most efficient protocol, Cheetah [14]. Recall that the plaintext of an AHE scheme is a polynomial, which can pack a large number of inputs to amortize the overhead [29, 13]. The key contribution of our proposed protocol is a more compact input packing approach.

Our starting point is that polynomial multiplication implies vector inner product, if we arrange the coefficients properly [14]. As shown in Equation 1, when multiplying two polynomials of degree-$(N$-$1)$, the $(N$-$1)$-th coefficient of the resulting polynomial is the inner product of the two coefficient vectors in opposite orders. By using an appropriate arrangement of coefficients, this idea can be extended to matrix multiplications, since they consist of a set of inner products. We give the definitions of two input packing functions $\pi_{\text{L}} : \mathbb{Z}_{2^\ell}^{m \times n} \to \mathbb{A}_{N, 2^\ell}$ and $\pi_{\text{R}} : \mathbb{Z}_{2^\ell}^{n \times k} \to \mathbb{A}_{N, 2^\ell}$ as follows:

$$\hat{x} = \pi_{\text{L}}(\mathbf{X}), \text{ s.t., } \hat{x}\,[i \cdot n \cdot k + (n-1) - j] = \mathbf{X}[i,j], \text{ for } i \in [m], j \in [n]$$
$$\hat{y} = \pi_{\text{R}}(\mathbf{Y}), \text{ s.t., } \hat{y}\,[j \cdot n + i] = \mathbf{Y}[i,j], \text{ for } i \in [n], j \in [k]$$

where all other coefficients of $\hat{x}$ and $\hat{y}$ are set to 0. Multiplication of polynomials $\hat{z} = \hat{x} \cdot \hat{y}$ directly gives the result of matrix multiplication $\mathbf{Z} = \mathbf{XY} \mod 2^\ell$ in some of $\hat{z}$'s coefficients. We formalize this process as below, and give a toy example in Figure 2.

**Theorem 4.1.** *Assuming $mnk \leq N$ and given two polynomials $\hat{x} = \pi_L(\mathbf{X})$ and $\hat{y} = \pi_R(\mathbf{Y})$, the matrix multiplication $\mathbf{Z} = \mathbf{XY} \mod 2^\ell$ can be evaluated via the product $\hat{z} = \hat{x} \cdot \hat{y}$, where $\mathbf{Z}[i,j]$ is computed in $\hat{z}\,[i \cdot n \cdot k + (j+1) \cdot n - 1]$ for $i \in [m]$ and $j \in [k]$.*

We defer the proof to Appendix A.2.1. When $mnk > N$, we first partition the matrices $\mathbf{X}, \mathbf{Y}$ into sub-matrices of $m_w \times n_w$ and $n_w \times k_w$ elements, respectively, such that $m_w n_w k_w \leq N$. Zero-padding is required when $m_w \nmid m$, $n_w \nmid n$ or $k_w \nmid k$. The protocol is shown in Algorithm 1.

**Complexity and security analysis**. For complexity, totally, two parties interact $\frac{k}{k_w}(\frac{m}{m_w} + \frac{n}{n_w})$ ciphertexts, and operate with $O(mnk/N)$ homomorphic additions and multiplications. We formalize the selection of parameters $m_w, n_w, k_w$ as an optimization problem, to minimize the communication

cost. Through our analysis in Appendix A.2.2, in general scenarios, our method theoretically achieves $\sqrt{m}$ communication improvement over Cheetah [14]. Notably, when $mnk \leq N$, we reduce the communication cost by a factor of $m$, since Cheetah encodes each row of the matrix into a ciphertext. Besides, the security proof is shown in Appendix A.2.3.

---

**Algorithm 1** Secure Matrix Multiplication Protocol

---

**Input:** $P_0$ holds $\mathbf{X} \in \mathbf{Z}_{2^\ell}^{m \times n}$, and $P_1$ holds $\mathbf{Y} \in \mathsf{Z}_{2^\ell}^{n \times k}$.

**Output:** $P_0$ and $P_1$ get $\langle \mathbf{Z} \rangle_0, \langle \mathbf{Z} \rangle_1 \in \mathbb{Z}_{2^\ell}^{m \times k}$, respectively, where $\mathbf{Z} = \mathbf{XY}$.

1: $P_0, P_1$ compute the partition window size $0 < m_\mathrm{w} \leq m, 0 < n_\mathrm{w} \leq n$ and $0 < k_\mathrm{w} \leq k$ such that $m_\mathrm{w} n_\mathrm{w} k_\mathrm{w} \leq N$, and set $n' = \lceil n/n_\mathrm{w} \rceil$, $m' = \lceil m/m_\mathrm{w} \rceil$, and $k' = \lceil k/k_\mathrm{w} \rceil$.

2: $P_1$ partitions the matrix $\mathbf{Y}$ into block matrices $\mathbf{Y}_{\beta,\gamma} \in \mathbb{Z}_{2^\ell}^{n_\mathrm{w} \times k_\mathrm{w}}$ for $\beta \in [n']$ and $\gamma \in [k']$.

3: $P_1$ encodes the matrices to polynomials $\hat{y}_{\beta,\gamma} = \pi_R(\mathbf{Y}_{\beta,\gamma})$ for $\beta \in [n']$ and $\gamma \in [k']$. Then $P_1$ sends to $P_0$ the ciphertexts $\{\mathrm{CT}_{\beta,\gamma} = \mathsf{Enc}(\hat{y}_{\beta,\gamma})\}$.

4: $P_0$ partitions the matrix $\mathbf{X}$ into block matrices $\mathbf{X}_{\alpha,\beta} \in \mathbb{Z}_{2^\ell}^{m_\mathrm{w} \times n_\mathrm{w}}$ for $\alpha \in [m']$ and $\beta \in [n']$. $P_0$ encodes the matrices to polynomials $\hat{x}_{\alpha,\beta} = \pi_L(\mathbf{X}_{\alpha,\beta})$.

5: $P_0$ uniformly at random samples plaintext polynomials $\hat{r}_{\alpha,\gamma}$ for $\alpha \in [m']$ and $\gamma \in [k']$. $P_0$ decodes these polynomials to a random mask $\mathbf{R} \in \mathbb{Z}_{2^\ell}^{m \times k}$ according to Theorem 4.1[4].

6: On receiving the ciphertexts $\{\mathrm{CT}_{\beta,\gamma}\}$ from $P_1$, $P_0$ operates $\mathrm{CT}'_{\alpha,\gamma} = \boxplus_{\beta \in [n']}(\hat{x}_{\alpha,\beta} \boxtimes \mathrm{CT}_{\beta,\gamma}) \boxminus \hat{r}_{\alpha,\gamma}$ for $\alpha \in [m']$ and $\gamma \in [k']$. Then $P_0$ sends to $P_1$ the ciphertexts $\{\mathrm{CT}'_{\alpha,\gamma}\}$.

7: $P_0$ outputs $\mathbf{R} \bmod 2^\ell$ as the share $\langle \mathbf{Z} \rangle_0$.

8: On receiving the ciphertexts $\{\mathrm{CT}'_{\alpha,\gamma}\}$ from $P_0$, $P_1$ computes $\langle \hat{z}_{\alpha,\gamma} \rangle_1 = \mathsf{Dec}(\mathrm{CT}'_{\alpha,\gamma})$ that are decoded to $\langle \mathbf{Z} \rangle_1$ using the method in Theorem 4.1.

---

### 4.2 Protocols for Non-linear Functions

Our non-linear protocols rely on several underlying protocols from the state-of-the-art works [13, 17]. In Figure 3, we enumerate the inputs and outputs of these protocols and then use them as a black box (See Appendix A.3.2 for details). On this basis, we provide efficient protocols for Softmax, GELU, and LayerNorm with specific optimizations. Additional details and security analysis are shown in Appendices A.3.3 and A.3.4.

| **Multiply** $\Pi_{\mathsf{Mul_{OT}}}$ | **Compare** $\Pi_{\mathsf{CMP}}$ | **NegExp** $\Pi_{\mathsf{nExp}}$ | **RecipSqrt** $\Pi_{\mathsf{rSqrt}}$ | **Recip** $\Pi_{\mathsf{Recip}}$ |
|---|---|---|---|---|
| Input: | Input: | Input: | Input: | Input: |
| • $P_b{:}\langle x \rangle_b, \langle y \rangle_b$ | • $P_b{:}\langle x \rangle_b$ | • $P_b{:}\langle x \rangle_b$ | • $P_b{:}\langle x \rangle_b$ | • $P_b{:}\langle x \rangle_b$ |
| Output: | Output: | Output: | Output: | Output: |
| • $P_b{:}\langle z \rangle_b$ | • $P_b{:}\langle z \rangle_b^B$ | • $P_b{:}\langle z \rangle_b$ | • $P_b{:}\langle z \rangle_b$ | • $P_b{:}\langle z \rangle_b$ |
| s.t. $z = xy$ | s.t. $z = 1\{x > 0\}$ | s.t. $z = e^x, x{<}0$ | s.t. $z = \frac{1}{\sqrt{x}}, x{>}\epsilon$ | s.t. $z = 1/x$ |

Figure 3: Underlying protocols from [13, 17]

### 4.2.1 Softmax

To evaluate attention layers, we need an efficient protocol to compute Softmax on secret-shared values. In particular, for a vector $\mathbf{x} \in \mathbb{Z}_{2^\ell}^d$, the Softmax function is denoted as $\mathsf{Softmax}_i(\mathbf{x}) = e^{x_i} / \sum_{j \in [d]} e^{x_j}$ for $i \in [d]$. The main challenge is to efficiently compute the underlying exponential function. Following the idea from [11, 30], we first normalize the input vector by $\mathbf{x} - \max_{i \in [d]} x_i$, which is always negative, and then invoke the existing exponential protocol $\Pi_{\mathsf{nExp}}$ in Figure 3 that only evaluates exponentiation on negative inputs. A simple analysis shows $\mathsf{softmax}(\mathbf{x} - \max_{i \in [d]} x_i)$ is equal to $\mathsf{softmax}(\mathbf{x})$. We evaluate max using a tree-reduction protocol, denoted by $\Pi_{\max}$. Specifically, we arrange the vector $\mathbf{x} \in \mathbb{Z}_{2^\ell}^d$ into a 2-ary tree with the depth of $\log d$, and evaluate the tree in a top-down fashion [31]. In each comparison of two secret-shared elements $x_i$ and $x_j$, we reduce it to the invocations of $\Pi_{\mathsf{CMP}}$ and $\Pi_{\mathsf{Mul_{OT}}}$, i.e., $\max(x_i, x_j) = \Pi_{\mathsf{Mul_{OT}}}(x_i - x_j, \Pi_{\mathsf{CMP}}(x_i - x_j)) + x_j$. With the above insight, the Softmax protocol is detailed in Algorithm 2.

SIRNN [17] also provides a solution for the generic exponential protocol by extending $\Pi_{\mathsf{nExp}}$. The idea is that the exponential of $x$ equals $\frac{1}{\mathsf{nExp}(-x)}$ if $x \geq 0$, and $\mathsf{nExp}(x)$ otherwise, where $\mathsf{nExp}$ is the

---

[4]This operation aims to avoid extra information leakage from the output polynomial's coefficients. The details refer to [14] and their implementations.

same as the exponential function except for negative inputs. Compared to our solution, realizing the softmax function with this generic exponential protocol additionally requires $d$ calls to the reciprocal protocol $\Pi_{\text{Recip}}$ and multiplication protocol $\Pi_{\text{MUT}_{\text{OT}}}$. Besides, compared with the generic library MP-SPDZ [16] that provides native support for exponentiation, our protocols achieve orders of magnitude improvement as shown in Section 5.2.

---

**Algorithm 2** Secure Softmax Protocol

---

**Input:** $P_0$, $P_1$ hold $\langle \mathbf{x} \rangle_0 \in \mathbb{Z}_{2^\ell}^d$, $\langle \mathbf{x} \rangle_1 \in \mathbb{Z}_{2^\ell}^d$, respectively.
**Output:** $P_0$, $P_1$ get $\langle \mathbf{y} \rangle_0 \in \mathbb{Z}_{2^\ell}^d$, $\langle \mathbf{y} \rangle_1 \in \mathbb{Z}_{2^\ell}^d$, respectively, where $\mathbf{y} = \text{Softmax}(\mathbf{x})$.
1: $P_0$, $P_1$ invoke $\Pi_{\max}(\mathbf{x})$ to compute $\langle \max(\mathbf{x}) \rangle$, where $\max(\mathbf{x}) = \max_{i \in [d]} x_i$.
2: For $i \in [d]$, $P_0$, $P_1$ invoke $\Pi_{\text{nExp}}$ on input $\langle \bar{x}_i \rangle$, and learn $\langle e^{\bar{x}_i} \rangle$, where $\bar{x}_i = x_i - \max(\mathbf{x})$.
3: $P_0$, $P_1$ invoke $\Pi_{\text{Recip}}$ with inputs $\langle \sum_{i \in [d]} e^{\bar{x}_i} \rangle$ and learn $\langle 1 / \sum_{i \in [d]} e^{\bar{x}_i} \rangle$.
4: For $i \in [d]$, $P_0$, $P_1$ invoke $\Pi_{\text{Mul}_{\text{OT}}}$ with inputs $\langle 1 / \sum_{i \in [d]} e^{\bar{x}_i} \rangle$ and $\langle e^{\bar{x}_i} \rangle$, and set outputs as $\langle y_i \rangle$.

---

#### 4.2.2 GELU

Rather than crypto-friendly ReLU [13], Transformer-based models utilize GELU activations [15], which can be represented as $\text{GELU}(x) = 0.5x \left( 1 + \text{Tanh} \left[ \sqrt{2/\pi} \left( x + 0.044715x^3 \right) \right] \right)$. The complete protocol is shown in Algorithm 3, in which we provide two insights to reduce the cost of GELU. First, we present an optimized protocol for the square of a secret-shared input $\langle x \rangle$. This relies on the observation: $x^2 = \langle x \rangle_0^2 + \langle x \rangle_1^2 + 2\langle x \rangle_0 \langle x \rangle_1$, where the first two terms can be locally computed by $P_0$ and $P_1$. We only invoke OT protocols to compute the cross term $2\langle x \rangle_0 \langle x \rangle_1$, and the optimized overhead is half that of the multiplication protocol $\Pi_{\text{Mul}_{\text{OT}}}$.

Second, we further optimize the evaluation of $\text{Tanh}(x) = \frac{e^{2x} - 1}{e^{2x} + 1}$. We observe that the sign of $x$ is equal to the sign of $\text{Tanh}(x)$, which allows us to leverage the negative exponential protocol $\Pi_{\text{nExp}}$ almost for free. At a high level, our Tanh protocol first learns the sign of the input $x$, and then evaluates Tanh on the negative input $\bar{x}$ with the constraint $|\bar{x}| = |x|$. Finally, the protocol generates the real output $\text{Tanh}(x)$ that equals $\text{Tanh}(\bar{x})$ if $x \leq 0$, and $-\text{Tanh}(\bar{x})$ otherwise. Our leaner Tanh protocol is given in Algorithm 5 of Appendix A.3.1. SIRNN [17] recently proposed the most efficient Tanh protocol with the insight of $\text{Tanh}(x) = 2\text{Sigmoid}(2x) - 1$, where $\text{Sigmoid}(x) = \frac{1}{1 + e^{-x}}$. However, the evaluation of Sigmoid uses the same idea as the general exponential protocol, as shown in Section 4.2.1. Compared with the protocol in SIRNN, our recipe for Tanh saves one invocation of the reciprocal protocol $\Pi_{\text{Recip}}$ and multiplication protocol $\Pi_{\text{Mul}_{\text{OT}}}$.

---

**Algorithm 3** Secure GELU Protocol

---

**Input:** $P_0$, $P_1$ hold $\langle x \rangle_0 \in \mathbb{Z}_{2^\ell}$, $\langle x \rangle_1 \in \mathbb{Z}_{2^\ell}$, respectively.
**Output:** $P_0$, $P_1$ get $\langle y \rangle_0 \in \mathbb{Z}_{2^\ell}$, $\langle y \rangle_1 \in \mathbb{Z}_{2^\ell}$, respectively, where $y = \text{GELU}(x)$.
1: $P_0$, $P_1$ invoke $\Pi_{\text{Mul}_{\text{OT}}}$ with inputs $\langle x \rangle$, and set their outputs as $\langle z \rangle = \text{Fix}(\sqrt{2/\pi})(\langle x \rangle + \text{Fix}(0.044715)\langle x \rangle^3)$
2: $P_0$, $P_1$ invoke $\Pi_{\text{Tanh}}$ with inputs $\langle z \rangle$, and set their outputs as $\langle \text{Tanh}(z) \rangle$.
3: $P_0$, $P_1$ invoke $\Pi_{\text{Mul}_{\text{OT}}}$ with inputs $\langle \text{Fix}(0.5)x \rangle$ and $\langle \text{Fix}(1) + \text{Tanh}(z) \rangle$, and output $\langle y \rangle$.

---

#### 4.2.3 LayerNorm

For a vector $\mathbf{x} \in \mathbb{Z}_{2^\ell}^d$, the LayerNorm function is denoted by $\text{LayerNorm}_i(\mathbf{x}) = \gamma(x_i - \mu)/\sigma + \beta$ for $i \in [d]$, where $\mu = \sum_{i \in [d]} x_i / d$ and $\sigma = \sqrt{\sum_{i \in [d]} (x_i - \mu)^2}$. In contrast to batch normalization (BN) in CNNs that is evaluated for free [14], LayerNorm requires multiplication and reciprocal square root operations. In our implementation, we observe that the multiplication dominates the overhead of LayerNorm. To address this issue, we adopt the same idea in GELU to compute the square of $x_i - \mu$, which saves half of the communication and computation costs.

Besides, inspired by the optimization in BN [14], we apply a LayerNorm merge technique to further reduce the overhead. Specifically, we make the observation that the weights $\gamma$ and $\beta$ of the LayerNorm layer are already known by $P_0$. As a result, $P_0$ first multiplies the scale factor $\gamma$ with the weights of

Table 1: Comparing the runtime (sec) and communication (MB) costs of our matrix multiplication and non-linear protocols with SOTA

| Matrix Multiplication | | | | | | | Non-Linear Protocols | | | | | | |
|---|---|---|---|---|---|---|---|---|---|---|---|---|---|
| Methods | Dims=(32, 8, 16) | | (128, 64, 128) | | (128, 768, 768) | | Methods | Softmax | | LayerNorm | | GELU | |
| | Time | Comm. | Time | Comm. | Time | Comm. | | Time | Comm. | Time | Comm. | Time | Comm. |
| Ours | 0.006 | 0.11 | 0.066 | 1.74 | 1.71 | 15.45 | Ours | 4.78 | 206.265 | 2.34 | 102.435 | 0.30 | 10.07 |
| Cheetah | 0.16 | 2.79 | 0.77 | 14.78 | 6.10 | 134.37 | SIRNN | 7.95 | 347.71 | 4.16 | 184.42 | 0.38 | 14.07 |
| | (26×) | (25×) | (11×) | (8×) | (3×) | (8×) | | (1.7×) | (1.7×) | (1.8×) | (1.8×) | (1.3×) | (1.4×) |
| SIRNN | 0.04 | 1.34 | 1.59 | 70.08 | 110.33 | 4920.08 | MP-SPDZ | 297.75 | 172,837 | 202.75 | 101,642 | 15.34 | 7,908.69 |
| | (6×) | (12×) | (23×) | (40×) | (64×) | (318×) | | (62×) | (837×) | (86×) | (992×) | (51×) | (785×) |

the next linear layer. After invoking the linear layer protocol on the scaled weights, $P_0$ adds the shift weight $\sigma$ to his additive share. The optimized protocol for LayerNorm is presented in Algorithm 4.

---

**Algorithm 4** Secure LayerNorm Protocol

---

**Input:** $P_0, P_1$ hold $\langle \mathbf{x} \rangle_0 \in \mathbb{Z}_{2^\ell}^d$, $\langle \mathbf{x} \rangle_1 \in \mathbb{Z}_{2^\ell}^d$, respectively.
**Output:** $P_0, P_1$ get $\langle \mathbf{y} \rangle_0$, $\langle \mathbf{y} \rangle_1$, respectively, where $\mathbf{y} = \text{LayerNorm}(\mathbf{x})$.
 1: For $i \in [d]$, $P_0, P_1$ invoke $\Pi_{\text{Mul}_{\text{OT}}}$ to compute $\langle (x_i - \mu)^2 \rangle$, where $\mu = \sum_{i \in [d]} x_i / d$.
 2: $P_0, P_1$ invoke $\Pi_{\text{rSqrt}}$ with inputs $\langle \sum_{i \in [d]} (x_i - \mu)^2 \rangle$ to learn output $\langle \frac{1}{\sigma} \rangle$.
 3: For $i \in [d]$, $P_0, P_1$ invoke $\Pi_{\text{Mul}_{\text{OT}}}$ with inputs $\langle \frac{1}{\sigma} \rangle$ and $\langle x_i - \mu \rangle$, and set outputs as $\langle y_i \rangle$.

---

## 5  Evaluation

### 5.1  Experimental Setup

**Implementation.** `Iron` is built on top of the SEAL library [32] and the EMP toolkit [33] in C++. We also use the EzPC framework [34]. This framework compiles a high-level TensorFlow code to secure computation protocols, which are then executed by our designed cryptographic backends. Like [17], we simulate a LAN network setting, where the bandwidth is 377 MBps and the echo latency is 0.8ms. All the following experiments are performed on AWS c5.9xlarge instances with Intel Xeon 8000 series CPUs at 3.6GHz.

**Datasets and Models.** We evaluate `Iron` on four NLP models from [35, 36]: BERT-Tiny, BERT-Medium, BERT-Base and BERT-Large. These models are parameterized by three hyper-parameters: the number of blocks, the dimension of representations and the number of input tokens (refer to Appendix A.4.1 for the hyper-parameters of these models). We train the models for four NLP tasks over the datasets of the Stanford Sentiment Treebank (SST-2), the Microsoft Research Paraphrase Corpus (MRPC), the Multi-Genre Natural Language Inference Corpus (MNLI) and the Stanford Question Answering Dataset (QNLI) from GLUE benchmarks [18].

### 5.2  Microbenchmark Evaluation

**Matrix multiplication**. In the left part of Table 1, we compare the performance of the proposed matrix multiplication protocol with the state-of-the-art counterparts of Cheetah [14] and SIRNN [17]. For fairness, we follow Cheetah for the parameter setup of homomorphic encryption. Compared with Cheetah, our rumtime is $3 \sim 26\times$ faster, and our communication cost is $8 \sim 25\times$ lower, depending on the input size. Notably, for small-size matrices (e.g., ones with $32 \times 8$ and $8 \times 16$ elements), our protocol only requires 0.11 MB communication and 6 ms runtime, while Cheetah achieves the computation with 2.79 MB and 160 ms. The reason is that our protocol encrypts the whole matrix into a single ciphertext, while Cheetah could only encrypt each row into a ciphertext, totally $m$ ciphertexts ($m$ is the number of rows of the output matrix). Moreover, compared with SIRNN that implements the most efficient OT-based matrix multiplication protocol, our protocol incurs up to two orders of magnitude less communication and an order of magnitude less runtime.

**Non-linear functions**. The right part of Table 1 shows the comparison of our Softmax, GELU and LayerNorm protocols with the generic MP-SPDZ framework [16] and the state-of-the-art SIRNN library [17] for math functions. It is worth noting that we implement some functions that these

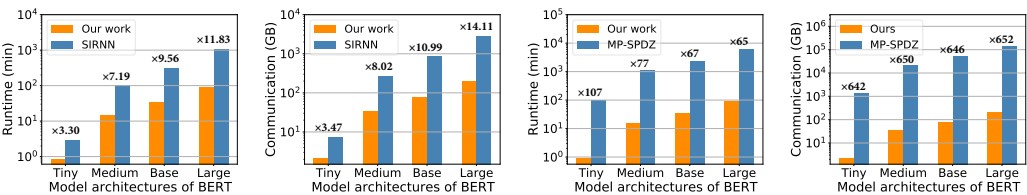

Figure 4: End-to-end comparisons with SIRNN and MP-SPDZ

frameworks did not provide before. In particular, GELU and LayerNorm are implemented in MP-SPDZ by calling its built-in functions for tanh, square root, and reciprocal, while we add Softmax and GELU protocols in SIRNN utilizing sigmoid and exponent functions. As shown in this table, our protocols are orders of magnitude better than those of MP-SPDZ, in terms of both runtime and communication. In particular, for the communication cost, our protocols achieve $785 \sim 993\times$ improvement. Moreover, while our protocols are built upon the underlying protocols from SIRNN, we also achieve $1.3 \sim 1.8\times$ lower runtime and $1.4 \sim 1.8\times$ lower communication due to our customized optimizations. Such an improvement gives us significant savings in communication complexity, since the non-linear layer protocols dominate the overall overhead, as described below.

## 5.3 End-to-end Inference Evaluation

**Comparison with prior methods.** In the left part of Figure 4, we evaluate our protocols on 4 BERT models compared with SIRNN [17]. It is observed that our rumtime is $3.3 \sim 11.83\times$ faster than that of SIRNN, and our communication cost is $3.47 \sim 14.11\times$ lower over four models. Moreover, our performance gains scale up as the model size grows. This is because our protocols achieve better amortized overhead when processing large-scale evaluations. We also compare the end-to-end private inference with MP-SPDZ in Figure 4. The results show that our protocols are orders of magnitude better, in terms of both time and communication costs. This is because specialized protocols are more communication efficient than generic alternatives, which is also observed by SIRNN.

**Performance breakdown.** In Figure 5, we present the runtime and communication breakdown of `Iron` on four BERT models. For clarity, we just report the result of one encoder. Recall that our private inference can be divided into linear protocols and non-linear protocols. For linear protocols, we observe that as the model size increases, the proportion of communication overhead remains approximately constant, accounting for $12 \sim 13\%$. This shows that our compact ciphertext encoding method effectively reduces the size of communication. For non-linear functions, as the model size increases, the proportion of computation overhead gradually decreases, from $84\%$ in BERT-Tiny to $76\%$ in BERT-Large. The savings come from amortizing the communication and computation costs by packing data from large tensors. Despite such advantages, the main bottleneck of our work is the communication overhead of non-linear layers. However, it is an open problem to solve the communication issue while maintaining the model accuracy.

**Accuracy comparison with plaintext.** In the left part of Figure 6, we show the accuacy of plaintext (float-point) and `Iron` (fixed-point) on the BERT-Tiny model. We observe that the accuracy achieved by `Iron` matches the accuracy of the plaintext TensorFlow code. Specifically, the accuracy loss does not exceed $0.3\%$ over all datasets, and surprisingly, `Iron` exceeds the plaintext baseline on MNLI by $0.85\%$. Similar results also appear in private CNN inference [13]. Such accuracy advantages experimentally validate that our protocols are numerically precise. Moreover, in the right part of Figure 6, we also compare the accuracy with the plaintext baseline, as the fractional scale varies on MRPC. We observe that `Iron` with the scale of 12 exactly matches the accuracy of the plaintext model. The accuracy loss is lower than $1\%$ when the scale $\geq 6$. This conclusion is in line with the prior work [13], namely that neural networks can tolerate stochastic fault behavior [37].

## 6 Discussion

We discuss possible solutions to further improve the efficiency of private inference on Transformers. They are compatible with our proposed protocols and hence can be directly integrated into our framework.

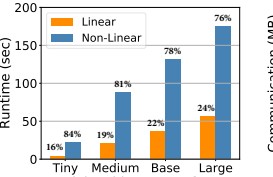 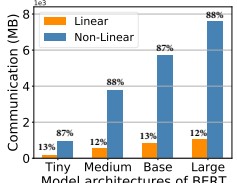 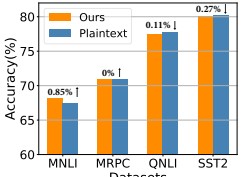 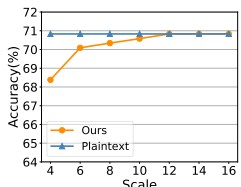

Figure 5: Performance breakdown on BERT.    Figure 6: Accuracy comparison with plaintext

**Pushing expensive operations into an offline phase.** Our framework is readily extended to the pre-processing model, including an offline client-input independent phase and an online input-dependent phase. This paradigm has been instantiated in private CNN inference [38, 8], and the results show that about 99% of the cryptographic overhead can be moved to the offline phase. We briefly outline how to build our protocols with this paradigm. For linear operations, we can generate in advance Beaver's triple in the matrix form [22, 38] using our matrix multiplication protocol. In the online phase, these triples are consumed, and it additionally requires non-cryptographic operations and plaintext communication. For non-linear layers, almost all operations crucially rely on the OT protocols, which can be pre-generated in the offline phase [39, 14]. Like linear protocols, the online phase is cheap.

**Using mixed-bitwidths in private inference.** `Iron` works with a uniform bitwidth, which is required to be large enough to accommodate all intermediate values. While effectively avoiding integer overflows, our protocols may cause intensive communication, since the performance of non-linear operators depends critically on bitwidths. Taking inspiration from [17, 40], one possible optimization is to employ non-uniform (mixed) bitwidths, operate in low bitwidths and switch to high bitwidths only when necessary. To this end, the mixed-precision model should be generated with proper compilers by using techniques like quantization [41, 42]. With mixed-bitwidths fixed-point models, our protocols can be integrated seamlessly.

**Applying orthogonal model optimizations.** We can improve performance by simplifying the model architecture such as model pruning and advanced neural architecture search [43, 44], which have been commonly adopted in private CNN inference [12, 45, 37]. Like prior works [12, 45, 37] that find and tailor models to the requirements of private inference, we can define a proper search space with the goal of decreasing the overhead, e.g., substituting costly Softmax and GELU functions, or reducing the matrices' dimension in matrix multiplication. As mentioned in Section 1, the concurrent work, THE-X [19], has explored replacing complex math functions with HE-friendly alternatives while achieving comparable accuracy.

## 7   Conclusion

We propose `Iron`, an efficient cryptographic framework for private Tramsformer inference. Specifically, we carefully design a new encoding method for optimizing homomorphic encryption-based matrix multiplication. Further, we devise several communication-efficient non-linear protocols like Softmax, LayerNorm and GELU by integrating advanced secret-sharing primitives. Experimental results show that `Iron` outperforms prior art by up to one order of magnitude in terms of computation and communication overheads. We believe that our novel protocols would help advance the practical instantiations of private Transformer inference.

## Acknowledgment

The authors would like to thank the anonymous reviewers for their insightful comments. This work is supported by the Key-Area Research and Development Program of Guangdong Province under Grant 2020B0101360001, National Natural Science Foundation of China under Grants 62020106013, 61972454, and 61802051, Sichuan Science and Technology Program under Grants 2020JDTD0007 and 2020YFG0298, the Fundamental Research Funds for Chinese Central Universities under Grant ZYGX2020ZB027, and Singapore Ministry of Education (MOE) AcRF Tier 2 MOE-T2EP20121-0006.

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
