# A Appendix

## A.1 More Details on Preliminaries

### A.1.1 Fixed-Point Encoding

Same as other neural networks, Transformer-based models use floating-point arithmetic, however cryptographic protocols operate on integers. Therefore, we require a float-to-integer conversion [46, 30, 17] to represent a floating-point number $x \in \mathbb{Q}$ into the ring $\mathbb{Z}_{2^\ell}$. Specifically, we first encode it as a fixed-point number, which is parametrized by a scale parameter $s$ that determines the fractional precision. Then, we embed the fixed-point representation into the ring with 2's complement representation. The formulation is $a = \lfloor 2^s \times x \rfloor \in \mathbb{Z}_{2^\ell}$ if $x$ is a non-negative number, and $a = 2^\ell - \lfloor 2^s \times |x| \rfloor \in \mathbb{Z}_{2^\ell}$ if $x$ is a negative number, where $s$ is the length of the (binary) fractional bits and $\ell$ is the bitwidth of the secret sharing ring. Unless otherwise stated, similar as prior works [14], we set the bitwidth as 37 and the scale as 12 in the fixed-point encoding. Because of the use of the above fixed-point encoding, after multiplication, the scale of the output is $2s$. Therefore, a truncation operation is required to reduce scale. We use the $\Pi_{\text{Trunc}}$ protocol proposed in [13] and improved by [17], which leads to faithful implementation of fixed-point arithmetic. For simplicity, we omit this operation in our protocol description. The overhead of truncation in `Iron` will be reported in Table 4.

### A.1.2 Formal Description of the Threat Model

Same as prior private inference works [13, 14, 17], the security of `Iron` is provably provided in the simulation paradigm against static honest-but-curious probabilistic polynomial-time (PPT) adversaries. Namely, a PPT adversary $\mathcal{A}$ passively corrupts either the server or the client at the beginning of the protocol and honestly follows the protocol specification. In the simulation paradigm, two worlds are defined: a real world where the server and the client perform the protocol according to the specification in the presence of $\mathcal{A}$, and an ideal world where the parties send their inputs to a trusted dealer (also called functionality) that executes the evaluation faithfully. The executions in both worlds are coordinated by the environment Env, which chooses the inputs to the parties and plays the role of a distinguisher between the real and ideal executions. It is required that for any adversary, the real-world distribution is computationally indistinguishable to the ideal-world distribution. Some of our protocols invoke sub-protocols and we describe them using the hybrid model. This is similar to a real execution, except that sub-protocols are replaced by the invocations of the corresponding functionality instances. We recap the definition of a private inference protocol in [38] [14] as below.

**Definition A.1.** A protocol $\Pi_{\text{PI}}$ between the server having as input a model $M$ with weights $w$ and the client having as input a sample $x$ is a private inference protocol against honest-but-curious adversaries if it satisfies the following guarantees: 1) *Correctness*: on every model weights $w$ and every input sample $x$, the output of the client at the end of the protocol is the correct inference $M(w, x)$. 2) *Security*: For a corrupted client, there exists an efficient simulator $\text{Sim}_C$ such that $\text{View}_C^{\text{PI}}$ is computationally indistinguishable $\text{Sim}_C(\text{output})$, where $\text{View}_C^{\text{PI}}$ is the view of the client in the execution of $\Pi_{\text{PI}}$ and $\text{output}$ denotes the output of the inference. Similarly, for a corrupted server, there exists an efficient simulator $\text{Sim}_S$ such that $\text{View}_S^{\text{PI}}$ is computationally indistinguishable $\text{Sim}_S$.

Notice that the honest-but-curious security proof of `Iron` according to the above definition will follow trivially from sequential composibility of individual sub-protocols [47, 13, 30]. Hence, we require to provide a security proof for our matrix multiplication and non-linear protocols. We refer to Section A.2.3 and A.3.4 for the sub-protocols' security analysis.

### A.1.3 Multi-Head Attention

Instead of performing a single attention function, existing Transformer-based models [1, 2] follow a multi-head attention variant, which can be represented as

$$\text{MultiHeadAtten} = \text{Concat}\left(\text{Attention}(\mathbf{X}_{Q,j}, \mathbf{X}_{K,j}, \mathbf{X}_{V,j}), j \in [H]\right)\mathbf{W}_O, \tag{4}$$

where $H$ is the number of heads, and $\mathbf{X}_{Q,j} = \mathbf{X}_Q\mathbf{W}_{Q,j}$, $\mathbf{X}_{K,j} = \mathbf{X}_K\mathbf{W}_{K,j}$, $\mathbf{X}_{V,j} = \mathbf{X}_V\mathbf{W}_{V,j}$ for $j \in [H]$. The main intuition is that multi-head attention allows the model to jointly attend to information from different representation subspaces at different positions [1].

## A.2 More Details on the Matrix Multiplication Protocol

### A.2.1 Correction Proof of the Matrix Multiplication Protocol

*Proof.* For each $i \in [m]$ and $j \in [k]$, we write $\epsilon_{i,j} = i \cdot n \cdot k + (j+1) \cdot n - 1$ for simplicity. Based on the description of Section 4.1, for $\epsilon_{i,j} \geq nk$, $\hat{y}[\epsilon_{i,j}] = 0$ holds. Therefore, given the definition of Equation 1, we have $\hat{z}[\epsilon_{i,j}] = \sum_{0 \leq \mu < n} \hat{x}[i \cdot n \cdot k + (n-1) - \mu]\hat{y}[j \cdot n + \mu] = \sum_{0 \leq \mu < n} \mathbf{X}[i, \mu]\mathbf{Y}[\mu, j]$, which is exactly $\mathbf{Z}[i][j]$. $\qquad\square$

### A.2.2 Optimal Parameters Selection in the Matrix Multiplication Protocol

As shown in Section 4.1, the matrix multiplication $\mathbf{X}_{m \times n} \cdot \mathbf{Y}_{n \times k}$ requires the communication of $\frac{m}{m_w}(\frac{n}{n_w} + \frac{k}{k_w})$ ciphertexts. To minimize the ciphertext communication cost, we formalize the selection of the parameters $m_w, n_w, k_w$ as an optimization problem, i.e., $\min_{\{m_w, n_w, k_w\}} \frac{m}{m_w}(\frac{n}{n_w} + \frac{k}{k_w})$, s.t., $m_w n_w k_w \leq N$, where $m, n, k, N$ are constants. Given the difficulty of solving the above multivariate optimization, we figure out a sub-optimal solution. To this end, we first fix $m_w = m$[5], and hence the constraint is transformed to $n_w k_w \leq \frac{N}{m}$. Correspondingly, the new optimization problem is $\min_{\{n_w, k_w\}} \frac{n}{n_w} + \frac{k}{k_w}$, s.t., $n_w k_w \leq \frac{N}{m}$. Then, the following holds:

$$\frac{n}{n_w} + \frac{k}{k_w} = 1/\frac{n_w}{n} + 1/\frac{k_w}{k} \geq \frac{2\sqrt{nk}}{\sqrt{n_w k_w}} \geq \frac{2\sqrt{mnk}}{\sqrt{N}}, \tag{5}$$

where the first inequality is due to $\frac{1}{a} + \frac{1}{b} \geq \frac{2}{\sqrt{ab}}$, and the second inequality comes from $n_w k_w \leq \frac{N}{m}$. As a result, assuming $m_w = m$, the optimal communication size is $\frac{2\sqrt{mnk}}{\sqrt{N}}$ ciphertexts. Notably, it may not have the optimal analytical solution, because the variables must be discrete positive integers, rather than real number. To achieve minimal communication in our implementation, like Cheetah [14], we use the exhaustive testing approach on all the results satisfying the constraint to find the optimal matrix partitioning strategy. Note that this is quite fast due to the small search range.

**Communication comparison with Cheetah.** As shown in Section 3.1 of [14], the Cheetah's communication overhead is $m(\frac{n}{n_w} + \frac{k}{k_w})$ ciphertexts with the constraint $n_w k_w \leq N$. By using the similar analysis as above, we can obtain its optimal solution, i.e., $\frac{2m\sqrt{nk}}{\sqrt{N}}$. Therefore, we obtain a $\sqrt{m} \times$ communication improvement in an ideal situation (i.e., the optimal analytical integer solution exists).

### A.2.3 Security Proof of the Matrix Multiplication Protocol

**Theorem A.2.** *In presence of an honest-but-curious adversary, the protocol $\Pi_{\text{MatMul}}$ in Algorithm 1 realizes the matrix multiplication functionality, in which $P_0$ and $P_1$ take as inputs the matrices $\mathbf{X}$ and $\mathbf{Y}$, and learn the secret shares $\langle \mathbf{Z} \rangle_0$ and $\langle \mathbf{Z} \rangle_1$, respectively, such that $\mathbf{Z} = \mathbf{XY}$.*

*Proof.* The correctness of Theorem A.2 is directly derived from Theorem 4.1. We below focus on the protocol's security when the server or the client is corrupted. Our security proof follows the simulation paradigm defined in Section A.1.2. In this paradigm we need show that the real-world distribution is computationally indistinguishable to the simulated distribution by the simulator Sim in the ideal world.

**Proof of indistinguishability with the corrupted server.** The server's view of $\text{View}_S^{\text{MatMul}}$ consists of ciphertexts $\text{CT}_{\beta,\gamma}$. The simulator $\text{Sim}_S$ for this view can be constructed as follows:

*Given the access to public parameters, $\text{Sim}_S$ outputs ciphertexts $\widetilde{\text{CT}}_{\beta,\gamma} = \text{Enc}(0)$ to the server.*

The security against the corrupted server is directly reduced to the semantic security of the underlying homomorphic encryption scheme. Thus we have that the simulated view $\text{View}_S^{\text{MatMul}}$ in the ideal world is computationally indistinguishable from the real-world distribution of the protocol.

**Proof of indistinguishability with corrupted clients.** The client's view of $\text{View}_C^{\text{MatMul}}$ consists of ciphertexts $\text{CT}'_{\alpha,\gamma}$, and the decryption of these ciphertexts, i.e., $\langle \mathbf{Z} \rangle_1$. The simulator $\text{Sim}_C$ for this view can be constructed as follows:

---

[5] In our setting, $m$ is always less than $N$.

*On receiving the ciphertexts $\mathrm{CT}_{\alpha,\gamma}$ from the client, $\mathsf{Sim}_C$ samples uniformly random polynomials $\hat{r}_{\alpha,\gamma} \in \mathbb{A}_{N,2^\ell}$, and computes $\widetilde{\mathrm{CT}}'_{\alpha,\gamma} = \mathsf{Enc}(\hat{r}_{\alpha,\gamma})$. Given the access to the output, $\mathsf{Sim}_C$ outputs $\widetilde{\mathrm{CT}}'_{\alpha,\gamma}$ to the client.*

Similarly, the ciphertexts $\widetilde{\mathrm{CT}}'_{\alpha,\gamma}$ are computationally indistinguishable from $\widetilde{\mathrm{CT}}_{\alpha,\gamma}$ due to the semantic security. Besides, the values of $\langle \mathbf{Z} \rangle_1$ distribute uniformly in $\mathbb{Z}_{2^\ell}$, which is exactly the same distribution of values decoded from $\hat{r}_{\alpha,\gamma}$. Thus we have that $\mathsf{View}_C^{\mathsf{MatMul}}$ is computationally indistinguishable from the real-world distribution of the protocol. $\qquad\square$

### A.3 More Details on Non-linear Protocols

#### A.3.1 Tanh

---

**Algorithm 5** Secure Tanh Protocol

---

**Input:** $P_0, P_1$ hold $\langle x \rangle_0 \in \mathbb{Z}_{2^\ell}$, $\langle x \rangle_1 \in \mathbb{Z}_{2^\ell}$, respectively.
**Output:** $P_0, P_1$ get $\langle y \rangle_0 \in \mathbb{Z}_{2^\ell}$, $\langle y \rangle_1 \in \mathbb{Z}_{2^\ell}$, respectively, where $y = \mathsf{Tanh}(x)$.
1: $P_0, P_1$ parse $\langle x \rangle_0 = \mathsf{msb}_0 \| a_0$ and $\langle x \rangle_1 = \mathsf{msb}_1 \| a_1$, and invoke $\Pi_{\mathsf{CMP}}$ to learn $\langle \mathsf{carry} \rangle_b^B$, where $\mathsf{carry} = 1\{a_0 + a_1 > 2^{\ell-1} - 1\}$, and the inputs are $2^{\ell-1} - a_0 - 1$ and $a_1$ from $P_0$ and $P_1$, respectively. For $b \in \{0, 1\}$, $P_b$ outputs $\langle \mathsf{MSB}(x) \rangle_b^B = \langle \mathsf{carry} \rangle_b^B \oplus \mathsf{msb}_b$.
2: $P_0, P_1$ invoke $\Pi_{\mathsf{Mul_{OT}}}$ with inputs $\langle 2x \rangle$ and $\langle \mathsf{MSB}(x) \rangle^B$, and set outputs as $\langle \bar{x} \rangle$, where $\bar{x} = 2x \cdot \mathsf{MSB}(x) - x$ that is always negative with the constraint of $|\bar{x}| = |x|$.
3: $P_0, P_1$ invoke $\Pi_{\mathsf{nExp}}$ with negative inputs $\langle 2\bar{x} \rangle$ and learn $\langle e^{2\bar{x}} \rangle$.
4: $P_0, P_1$ invoke $\Pi_{\mathsf{Recip}}$ with inputs $\langle e^{2\bar{x}} \rangle$, and learns $\langle \bar{y} \rangle$ where $\bar{y} = 1 - \frac{2}{e^{2\bar{x}}+1}$.
5: $P_0, P_1$ invoke an instance of $\Pi_{\mathsf{Mul_{OT}}}$ on input $\langle \mathsf{MSB}(x) \rangle^B$ and $\langle \bar{y} \rangle$, and learn $\langle y \rangle$, where $y = \bar{y} + \mathsf{MSB}(x) \cdot (-2\bar{y})$.

---

We present an optimized protocol for $\mathsf{Tanh}$ in Algorithm 5, which builds over the protocol used in [17]. Our optimization relies on the observation: when evaluating on $x$, the sign of $\mathsf{Tanh}(x)$ is the same as that of $x$. Compared with the protocol in [17], our alternative saves one invocation of the reciprocal protocol $\Pi_{\mathsf{Recip}}$ and multiplication protocol $\Pi_{\mathsf{Mul_{OT}}}$.

#### A.3.2 Underlying Protocols from [17, 13]

We outline the underlying protocols from existing works [17, 13], and the detailed implementation could be found in the corresponding papers.

- Multiplication ($\mathsf{Mul_{OT}}$): The OT-based multiplication protocol $\Pi_{\mathsf{Mul_{OT}}}$ takes as input $\langle x \rangle \in \{0, 1\}^\ell$ and $\langle y \rangle \in \{0, 1\}^\ell$ and returns $\langle z \rangle$ such that $z = xy$. The well-known technique is proposed by ABY [22] and optimized in [13]. Currently, the optimal solution invokes $2\text{-}\mathsf{COT}_i$ for $i \in \{1, \ldots, \ell\}$ requiring communication $\ell(\lambda + \frac{\ell+1}{2})$ bits with 2 rounds that is equivalent to $\ell$ instances of $\mathsf{COT}_{\frac{\ell+1}{2}}$. A variant of multiplication is multiplexer[6], which takes as input $\langle x \rangle^B \in \{0, 1\}$ and $\langle y \rangle \in \{0, 1\}^\ell$ and outputs $\langle z \rangle \in \{0, 1\}^\ell$ such that $z = y$ if $x = 1$ and 0 otherwise. The multiplexer protocol $\Pi_{\mathsf{Mux}}$ can be realized by 2 parallel calls of $2\text{-}\mathsf{COT}_\ell$ with communication $2(\lambda + \ell)$ bits and 2 rounds.

- Comparison (CMP): The comparison protocol $\Pi_{\mathsf{CMP}}$ takes as input $\langle x \rangle \in \{0, 1\}^\ell$, and returns $\langle z \rangle$ such that $z = 1\{x \geq 0\}$. Recently, [13] gave an efficient protocol for $\Pi_{\mathsf{CMP}}$ with communication less than $\lambda\ell + 14\ell$ bits with $\log \ell$ rounds.

- Exponential on negative inputs (nExp): The exponential protocol $\Pi_{\mathsf{nExp}}$ takes as input $x \in \{0, 1\}^\ell$, where $x \leq 0$, and returns $\langle z \rangle$ such that $z = e^x$. The protocol is proposed by [17], which invokes digit decomposition to generate small-length inputs and integrates the OT-based lookup table technique to compute exponential on the small-length inputs.

---

[6]In the protocol description, we treat these two types of multiplication indiscriminately, but we implement them using different techniques.

- Reciprocal of Square Root (rSqrt): The protocol of square root's reciprocal, $\Pi_{\mathrm{rSqrt}}$, takes as input $x \in \{0,1\}^\ell$ and returns $\langle z \rangle \in \{0,1\}^\ell$ such that $z = \frac{1}{\sqrt{x}}$. [17] proposed the state-of-the-art OT-based protocol, which relies on the Goldschmidt's algorithm [48] that iterates on an initial approximation.

- Reciprocal (Recip): The reciprocal protocol $\Pi_{\mathrm{Recip}}$ takes as input $x \in \{0,1\}^\ell$ and returns $\langle z \rangle \in \{0,1\}^\ell$ such that $z = 1/x$. The most efficient implementation is proposed in [17] with a similar idea as the protocol $\Pi_{\mathrm{rSqrt}}$.

### A.3.3 Extra Optimization

**MSB-known protocol optimization.** As pointed out by [17], 2PC protocols could be designed in a far more efficient way when the MSB of the inputs are known. In particular, we optimize truncation and OT-based multiplication in this case. For example, the MSB-known truncation protocol requires $O(\lambda(s+3))$ communication, instead of $O(\lambda(\ell+3))$, where $\lambda$ is the security parameter, $\ell$ is the bit length of the secret sharing ring and $s$ is the fractional scale.

We elaborate the optimization for the GELU protocol in Algorithm 3, and the same idea can also be used in the Softmax and LayerNorm protocols. For GELU, we fist compute the shares of $\mathrm{MSB}(x)$, instead of calculating it in the latter Tanh protocol, and then in the following sub-process, we use this knowledge to reduce overhead. Moreover, we further observe that the GELU protocol implies more MSB-known operations if proper computation order is considered. We rewrite the GELU formulation as below:

$$\mathrm{GELU}(x) = 0.5 \left( x + x \mathrm{Tanh}\left[ \sqrt{2/\pi} x \left( 1 + 0.044715 x^2 \right) \right] \right). \tag{6}$$

We observe that $1 + 0.044715 x^2$ and $x \mathrm{Tanh}\left[ \sqrt{2/\pi} x \left( 1 + 0.044715 x^2 \right) \right]$ are always non-negative, where the latter holds because the sign $x$ equals to that of $\mathrm{Tanh}\left[ \sqrt{2/\pi} x \left( 1 + 0.044715 x^2 \right) \right]$.

### A.3.4 Security Proof of Non-linear Protocols

Similar as the security of protocols in [13, 17], our protocols directly follow in the hybrid model. In particular, the security of the Softmax and GELU protocols are easy to see in $(\mathrm{CMP}, \mathrm{nExp}, \mathrm{Recip}, \mathrm{MUL}_{\mathrm{OT}})$-hybrid. Besides, the security of the LayerNorm protocol follows in $(\mathrm{rSqrt}, \mathrm{MUL}_{\mathrm{OT}})$-hybrid.

## A.4 More Details on Experimental Evaluation

### A.4.1 Additional Experimental Setup

Table 2: Models and hyper-parameters

| Models | #Params | Hyper-parameters | | |
|---|---|---|---|---|
| | | $b$ | $d$ | $t$ |
| BERT-Tiny | 4.4M | 2 | 128 | 128 |
| BERT-Medium | 41.7M | 8 | 512 | 128 |
| BERT-Base | 110.1M | 12 | 768 | 128 |
| BERT-Large | 340M | 24 | 1024 | 128 |

Table 3: Datasets and tasks description

| Datasets | #Train | #Test | Task | Domain |
|---|---|---|---|---|
| SST-2 | 67K | 872 | Single-sentence 2-classification | Movie reviews |
| MRPC | 3.7K | 408 | Sentence pair 2-class paraphrase | News |
| MNLI | 393K | 2K | Sentence pair 3-class inference | Misc. |
| QNLI | 105K | 2K | Sentence pair 2-class inference | Wikipedia |

We evaluation `Iron` on 4 widely used pre-trained BERT models with different hyper-parameters, as shown in Table 2. We denote the number of blocks as $b$, the dimension of representations as $d$, and the number of input tokens as $t$. We always fix the number of self-attention heads to $d/64$ and the size of feed-forward features to $4d$. The end-task models are obtained by stacking a linear classifier on top of the Transformer architectures with fine-tuning. We follow the default fine-tuning hyper-parameters in [35], e.g., batch size 32, learning rate $2 \times 10^{-5}$ and epoch 3. Notice that any hyper-parameters optimization during the training phase is compatible with our scheme. Besides, we use 4 datasets for different tasks from GLUE [18], which include the Stanford Sentiment Treebank

Table 4: Detailed performance breakdown of our protocols on BERT

| Models | Metrics | MatMul | Truncation | GELU | Softmax | LayerNorm | Total |
|--------|---------|--------|------------|------|---------|-----------|-------|
| BERT-Tiny | Runtime (Sec) | 1.54 | 2.61 | 14.65 | 5.04 | 2.40 | 26.24 |
| | Comm. (MB) | 29.99 | 108.66 | 642.38 | 214.01 | 99.02 | 1094.07 |
| BERT-Medium | Runtime (Sec) | 11.25 | 9.70 | 58.79 | 20.24 | 8.56 | 108.53 |
| | Comm. (MB) | 132.00 | 404.63 | 2565.53 | 856.05 | 374.53 | 4332.74 |
| BERT-Base | Runtime (Sec) | 22.12 | 14.87 | 88.08 | 30.31 | 13.05 | 168.43 |
| | Comm. (MB) | 197.68 | 626.94 | 3848.30 | 1284.08 | 575.23 | 6532.23 |
| BERT-Large | Runtime (Sec) | 36.66 | 19.50 | 117.45 | 40.43 | 16.65 | 230.70 |
| | Comm. (MB) | 240.05 | 809.25 | 5131.06 | 1712.10 | 733.83 | 8626.28 |

(SST-2), the Microsoft Research Paraphrase Corpus (MRPC), the Multi-Genre Natural Language Inference Corpus (MNLI) and the Stanford Question Answering Dataset (QNLI). Table 3 shows the datasets' details. Besides, we assume the embedding table is publicly available to all parties, and hence the evaluation does not include the results of embedding layers.

### A.4.2 Additional Experimental Results

**Detailed performance breakdown on BERT.** In Table 4, we show the detailed performance breakdown including the communication and computation costs of matrix multiplication, truncation, GELU, softmax and layer normalization. The most expensive non-linear operation is GELU due to its huge number. For example, for each layer of BERT-Base, the number of GELU is 393,216. We also observe that the our linear operation is lightweight in terms of communication.

### A.5 Related Works

Recently a quantity of works have designed customized protocols for performing private inference on neural networks, especially convolutional neural networks. These special-purpose protocols improve the computation and communication costs and generally fall into two categories: linear protocols and non-linear protocols. We briefly discuss the progress as below.

**Linear protocols.** Gazelle [29] proposed an optimized AHE-based linear algebra kernels, which support matrix-vector multiplication and convolution operations. The main innovation is a new packing method to minimize the expensive rotation operations, which is the critical component for the linear algebra. After that, CrypTFlow2 [13] proposed a comprehensive implementation for linear layers, based on both AHE-based and OT-based solution[7]. For the AHE-based solution, they use the protocol from Gazelle, and employ several optimizations such as parallelization and reducing ciphertext size. They observe the AHE-based solution performs better than the OT-based counterpart, especially for large-scale models. More recently, Huang et al. presented Cheetah [14], the most efficient AHE-based linear layer protocols, including matrix-vector multiplication and convolution operations. The improvement comes from a novel input packing technique, which is rotation-free and hence efficient. Moreover, the packing method is compatible with secret sharing in a ring. This support further benefits the subsequent non-linear operations [14]. However, existing protocols are only optimized for matrix-vector multiplication, rather than general matrix multiplication `Iron` relies on. As mentioned earlier, directly extending the most efficient matrix-vector multiplication protocol still causes prohibitively high communication overhead. Therefore, to approach such communication issue, we propose a special-purpose protocol for matrix multiplication, based on the state-of-the-art protocol in Cheetah.

Notice that different from the setting of 2PC private inference, [49] proposed a private *outsourced* inference scheme, which stands for encrypted data and encrypted model. To this end, [49] designs a homomorphic matrix multiplication protocol for multiplying two encrypted matrices, which is fundamentally different our homomorphic multiplication with a plaintext. As a result, it requires to invoke costly homomorphic multiplication and rotation operations, which are about $2 \sim 20\times$ more expensive than the underlying operations of our protocol (refer to Table 9 of [50]).

---

[7]The OT-based linear protocol is also used in SIRNN [17]

**Non-linear protocols.** Although earlier works [29, 38] implemented non-linear function evaluation with garbled circuits (GC), CrypTFlow2 [13] found that these GC-based solutions result in high communication overhead. Therefore, the authors designed optimized OT-based protocols, such as truncation and comparison. These protocols achieve state-of-the-art performance, and can be seen as general underlying building blocks for the design of advanced protocols [17]. Despite the efficiency advantage for truncation and comparison, these protocols can not support complex functions, like exponent in Transformers. Actually, the state-of-the-art general-purpose framework, MP-SPDZ, provides comprehensive protocols. However, as shown in SIRNN [17], the protocols implemented with MP-SPDZ are communication-heavy and computation-intensive. Therefore, SIRNN [17] proposed special-purpose protocols for exponent on negative inputs, sigmoid and reciprocal of square root, which achieve orders of magnitude improvement over MP-SPDZ, both in terms of runtime and communication. However, these functions are still insufficient to implement a private inference framework on Transformers. Therefore, on the basis of the building blocks in [13, 17], we propose new protocols for three non-linear functions that are critical components for Transformers, and make several specialized optimizations. Note that, [11] also proposed a softmax protocol but in an unrealistic setting, where except the client and the server, a trusted third party (TTP) exists and assists to generate correlated randomness to accelerate protocol evaluation. However, in a practical application, it is difficult to have a completely TTP [13, 14]. In contrast to [11], our setting lies in a practical client-server setting, without any unrealistic assumptions.