# OpenReview forum: "Iron: Private Inference on Transformers"
_NeurIPS.cc/2022/Conference — NeurIPS 2022 Accept_

### Official Review · Reviewer_6r55 · 2022-07-09

**Rating:** 6
**Confidence:** 5
**Soundness:** 3 good
**Presentation:** 2 fair
**Contribution:** 2 fair

**Summary:**

The paper proposes to use homomorphic encryption for Transformer model inference.  Moreover, the proposed method improves the speed of matrix operations. The experiments show that the inference results are numerically comparable.

**Questions:**

- (L267) Does the proposed method only support CPUs? There is no experiment conducted on GPUs.

- I did not find exact numbers of accuracy in the main content of the submission.

- Only NLP tasks with short input texts are evaluated. How about the performance and speed for longer inputs?

- How about the results on other tasks, such as vision problems?

- The speed is not directly compared with standard Transformer inference.

- How about the memory cost?

- What are the costs of private inference?

**Limitations:**

The paper can explicitly include discussions about the limitations.

**Strengths And Weaknesses:**

Strengths:

- Speedup the inference speed of homomorphic encryption for Transformer models.

- Because the Transformer network is hot now across research areas, how to host the models using private inference is worth exploring.

- The submission will also release the code?

==========

Weaknesses:

- The abstract claims "the first Transformer-based private inference framework". However, there was one paper https://aclanthology.org/2022.findings-acl.277/ that was published in ACL-22. The ACL-22 paper works on the same problem and also uses homomorphic encryption as the solution. The claim can be removed to make the claim more precise.

- (L267) Does the proposed method only support CPUs? There is no experiment conducted on GPUs. It makes the work less practical.

- (abstract) "which preserve the model accuracy of plaintext" The claim is not well supported by experiments. I did not find exact numbers of accuracy in the main content of the submission.

- Only NLP tasks with short input texts are evaluated. More applications can be demonstrated, especially for longer texts.

- The speed is not directly compared with standard Transformer inference. It seems that the inference is quite slow.

---

> ### Author Response · Authors · 2022-08-02
> **Response to Reviewer 6r55**
>
> Dear Reviewer,
>
> Thanks for your valuable comments. We address your questions below.
>
> ***Q1*. Concurrent work: THE-X: Privacy-Preserving Transformer Inference with Homomorphic Encryption. The claim "first soltion" can be removed to make the claim more precise.**
>
> *A1*. Thanks to the reviewer for bringing this concurrent work (THE-X) to our attention.
> THE-X and ours share the same goal, which is to enable private inference on Transformers with cryptographically provable security.
> However, we emphasize that this work was published after our submission.
> Below, we illustrate some important differences in terms of protocol design and security.
> We also have added the discussion in the related work of the revision and removed "the first" to make the claim more precise.
>
> 1) Protocol design. Our work aims to design new efficient protocols for the complex operations of Transformers, while orthogonal to ours, THE-X replaces them with HE-friendly operations.
> For example, THE-X replaces GELUs with simpler operations, i.e., ReLUs, and Softmax with the combination of ReLU and polynomials.
>
> 2) Security.
> Our work achieves more rigorous privacy protection than THE-X.
> Specifically, our work uses homomorphic encryption and secret sharing to securely evaluate all layers in ciphertext.
> As a result, the server learns nothing, while the client only obtains the prediction result in the private inference process.
> Such rigorous privacy guarantee is in line with recent state-of-the-art private inference works [2, 3, 4].
> However, in THE-X, the inputs of each non-linear layer are leaked to the client.
> Therefore, our work can be used to enhance the security of THE-X.
>
> ***Q2*. (abstract) "which preserve the model accuracy of plaintext" The claim is not well supported by experiments. I did not find exact numbers of accuracy in the main content of the submission.**
>
> *A2*. We apologize for the lack of accuracy in the main content.
> Due to the space limitation, we provided the accuracy results in the appendix of the submission.
> For convenience, we report the results in the table below, and additionally add the results on two vision tasks, except the NLP tasks we considered in the previous submission.
> We observe that the accuracy of private inference is comparable to that of plaintext, since our protocols are numerically precise.
> Moreover, slight changes are caused by the fixed-point (i.e., limited precision) representation in 2PC protocols.
> Interestingly, the accuracy of our private inference even outperforms the plaintext baseline on several datasets, which is also observed in private CNN infernce [3].
>
> |          |  MNLI  |  MRPC | QNLI | SST2 | CIFAR100 | ImageNet |
> | :---| :---: | :---: | :---: | :---: | :---: | :---: |
> | Plaintext | 67.35 | 70.83 | 77.72 | 80.16 | 82.87 | 78.26 |
> | Ours | 68.20 | 70.83 | 77.45 | 80.04 | 83.15 | 78.46 |
> | | | | | | | |
>
> ***Q3*. Only NLP tasks with short input texts are evaluated. More applications can be demonstrated, especially for longer texts. How about the performance and speed for longer inputs?**
>
> *A3*. To demonstrate our applications on longer texts, we add experiments on the SQuAD2.0 dataset.
> The text length is 384, which is greater than the size (i.e., 128) of the datasets we used previously.
> We report the communication and runtime overheads in the table below, and compare them with SIRNN [2].
> Experimental results show that we achieve $2.17 \times$ and $3.09 \times$ rumtime and communication improvements on SIRNN [2], respectively.
> This is in line with the advantages of the datasets used in the previous manuscript.
>
> We further emphasize that our work is readily extended to more models and tasks.
> This is because our protocols are designed for the generic operations of Transformers, rather than for specific models.
> Moreover, Transformer-based models share very similar architectures and operations, and hence they can be securely evaluated with our protocols.
>
> |          |  Time (Sec)  |  Comm. (GB)  |
> | :---| :---: | :---: |
> | SIRNN | 103.78 | 28.14 |
> | Ours| 47.85 | 9.10 |
> | | | |

---

> ### Author Response · Authors · 2022-08-02
> **Response to Reviewer 6r55**
>
> ***Q4*. How about the results on other tasks, such as vision problems?**
>
> *A4*. Thanks for your valuable suggestion.
> We add the evaluation of ViT on the CIFAR100 and ImageNet datasets, to further explore the scalability of our scheme.
>
> Specifically, we use the model architecture from [1] and the pre-trained weights from the open-sourced code [8].
> The architectures are 7-layer and 14-layer Transformers for CIFAR100 and ImageNet, respectively.
> In the following table, we report the overhead results of ours and SIRNN [2].
> It is observed that we achieve $2.70 \times \sim 3.05 \times$ and $4.40 \times \sim 5.62 \times$ improvement in terms of runtime and communication  costs, respectively.
> As shown in the above, the accuracy of CIFAR100 and ImageNet in private inference outperforms the plaintext accuracy by $0.28$\% and $0.2$\%, respectively.
>
> | CIFAR100 |  Time (Sec)  |  Comm. (GB)  |
> | :---| :---: | :---: |
> | SIRNN | 408.94 | 112.94 |
> | Ours| 151.03 | 25.64 |
> | | | |
>
> | ImageNet |  Time (Sec)  |  Comm. (GB)  |
> | :---| :---: | :---: |
> | SIRNN | 1271.82 | 381.17 |
> | Ours| 415.75 | 67.81 |
> | | | |
>
> ***Q5*. How about the memory cost?**
>
> *A5*. We test the memory cost during the evaluation of our private inference using the SAR (System Activity Report) tool [5].
> We evaluate on four Bert models, i.e., Bert-Tiny, Bert-Medium, Bert-Base and Bert-Large, and observe that the maximum memory consumption is $0.57$\% $\sim 6.40$\%.
>
> ***Q6*. Does the proposed method only support CPUs? There is no experiment conducted on GPUs. It makes the work less practical.**
>
> *A6*. We would like to clarify that our secure protocols are independent of specific hardwares, which can be evaluated on arbitrary GPUs and CPUs.
> However, we only support CPUs currently, mainly because the underlying cryptographic algorithms (e.g., homomorphic encryption) are implemented in CPUs.
> Therefore, our approach can be extended directly to use GPUs, once the corresponding GPU implementation of the underlying techniques is available.
> Besides, while several recent efforts have explored to accelerate private CNN inference with GPUs [6, 7], these works are tailored for specific cryptographic protocols.
> Undoubtedly, it is a promising future direction to design a general-purpose platform, which provides protocol-independent GPU acceleration, and apply it to our private Transformer inference.
>
> ***Q7*. The speed is not directly compared with standard Transformer inference. It seems that the inference is quite slow.**
>
> *A7*.The runtime of our work is indeed slower than that of the standard inference in plaintext.
> This performance gap also exists in recent state-of-the-art private inference works [2, 3, 4].
> For example, considering the private inference of ResNet32 on CIFAR100, the most advanced effort, Cheetah [4], still needs 15.95 seconds for each input data.
> As illustrated in [2], degradation of  performance is inevitable, once we aim to provide rigorous privacy protection (i.e., cryptographically provable security).
> The motivation of our work is to mitigate this gap in private Transformer inference.
> We design customized protocols for matrix multiplication and three non-linear functions, which outperform the state-of-the-art counterparts as shown in Section 5 of the manuscript.
> Furthermore, orthogonal works are to use model architecture optimizations and specific hardwares (e.g., GPUs) to further accelerate the inference process.
> We believe that our customized protocols would help advance the practical instantiations of private Transformer inference.
>
> ***Q8*. What are the costs of private inference?**
>
> *A8*. As demonstrated in [2], the costs of private inference mainly come from the computation and communication of the secure protocol for each operation.
>
> Specifically, we analysis the main cost of our private Transformer inference, in terms of linear layers and non-linear layers.
> For linear layers, instead of executing matrix multiplication on plaintext, our protocols need to 1) encryption and decryption operations, 2) communicating the encrypted matrices, and 3) performing matrix multiplication on ciphertexts.
> While our work proposes new solution and achieves reasonable performance, these encrypted operations and communication lead to expensive costs compared with standard operations.
> For non-linear layers, private inference requires invoking secret sharing protocols, which involve 1) multiple interactions and 2) additionally local operations.
> The communication cost dominates the overhead of these non-linear protocols.
> In conclusion, reducing the computational and communication complexity of the underlying protocols is important for putting private Transformer inference into practice.
>
> ***Q9*. The submission will also release the code?**
>
> *A9*. We have submitted our source code in the supplementary material in the revision, and will open source it in the near future.

---

> > ### Comment · Reviewer_6r55 · 2022-08-05
> > **question 7**
> >
> > ``Q7. The speed is not directly compared with standard Transformer inference. It seems that the inference is quite slow.
> >
> >
> > Thanks for your detailed response. For Q7, the comparisons between standard Transformer inference and HE inference can be explicitly added in the paper.

---

> > > ### Author Response · Authors · 2022-08-06
> > > **Response to question 7**
> > >
> > > Dear Reviewer,
> > >
> > > Thanks again for your valuable comments.
> > > We promise that we will add the comparisons between standard Transformer inference and private inference in the paper.

---

> ### Author Response · Authors · 2022-08-02
> **Response to Reviewer 6r55**
>
> References:
>
> [1] Escaping the Big Data Paradigm with Compact Transformers, arXiv, 2021.
>
> [2] SiRnn: A math library for secure RNN inference, IEEE S\&P, 2021.
>
> [3] CrypTFlow2: Practical 2-Party Secure Inference, ACM CCS, 2020.
>
> [4] Cheetah: Lean and Fast Secure Two-Party Deep Neural Network Inference, USENIX Security, 2022.
>
> [5] SAR: http://sebastien.godard.pagesperso-orange.fr/
>
> [6] CRYPTGPU: Fast Privacy-Preserving Machine Learning on the GPU, IEEE S\&P, 2021.
>
> [7] Piranha: A GPU Platform for Secure Computation, USENIX Secutiy, 2022.
>
> [8] CCT: https://github.com/SHI-Labs/Compact-Transformers

---

### Official Review · Reviewer_QBqi · 2022-07-11

**Rating:** 6
**Confidence:** 4
**Soundness:** 3 good
**Presentation:** 4 excellent
**Contribution:** 3 good

**Summary:**

The paper proposes a solution based on LHE+Secret Sharing for private inference of transformer models. The key challenge is in adapting different transformer layers, namely large matrix-matrix multiplication, Softmax, layer normalization, and GELU non-linear function.

The main contribution of this paper is an encoding scheme that is designed to make matrix-matrix multiplication in LHE more efficient than existing schemes, and a protocol for performing secure matrix multiplication. There are minor contributions in improving the efficiency of evaluating Softmax, layer normalization, and GELU.

Experiments are conducted on 4 NLP models, BERT-Tiny, BERTMedium, BERT-Base, and BERT-Large. Comparisons are performed against Cheetah, a baseline method in terms of runtime (latency) and communication efficiency.

**Questions:**

Overall I am positive about the paper, and will revise the score based on the author’s rebuttal.

- How does the proposed matrix-matrix multiplication scheme compare quantitatively and quantitatively to [1] above. And how do they all scale with matrix sizes since this is a key aspect of variation in transformers?

- Line 323-324 states, "For linear operations, we can generate in advance Beaver’s triple in the matrix form [21, 36] using our matrix multiplication protocol." Did you mean non-linear operations, unless I am missing something I am unsure why Beaver triples are needed for matrix multiplication?

**Limitations:**

There is no explicit discussion on the limitations of the current work. There is a separate section for discussing some orthogonal directions to augment the current work, in terms of mixed-bit widths and applying model optimizations.

**Strengths And Weaknesses:**

**Strengths:**
+ The paper is very well written in terms of providing the right amount of background, and clarity of descriptions. I enjoyed reading the paper.

+ Extending the concept of private inference for transformer-based models. Given the growing application of transformers, it is critical to design private inference-friendly versions of these models.

+  The speedup over SIRNN is notable.

+ The data encoding scheme and associated secure matrix multiplication protocol is effective.

**Weaknesses:**
- The primary contribution of the paper is a new data encoding scheme that is tailored for improving the efficiency of matrix-matrix multiplications. However, the paper does not compare (qualitatively or quantitatively) a key existing solution [1] for the same. The current paper only compares against the method in Cheetah, but it is not clear to this reviewer how it compares to [1]. I believe there is no single encoding scheme that is optimal for all matrix sizes.

- The paper appears to be an incremental improvement over CHEETAH. The matrix multiplication is certainly better, but other protocols are minor adaptations of existing practice (e.g., subtracting max before computing softmax is very common practice even in applications that focus only on plaintext) and those implemented in CHEETAH.

[1] Secure Outsourced Matrix Computation And Application to Neural Networks.

This is just a comment to make the authors aware of concurrent work [2] along the direction of adapting transformers for private inference with homomorphic encryption.
[2] THE-X: Privacy-Preserving Transformer Inference with Homomorphic Encryption, ACL 2022

---

> ### Author Response · Authors · 2022-08-02
> **Response to Reviewer QBqi**
>
> Dear Reviewer,
>
> Thanks for your valuable comments. We address your questions below.
>
> ***Q1*. How does the proposed matrix-matrix multiplication scheme compare qualitatively and quantitatively to [1]. How do they all scale with matrix sizes since this is a key aspect of variation in transformers?**
>
> *A1*. Thanks for your valuable comment! [1] designs a novel homomorphic algorithm for multiplying two encrypted matrices, and achieves the state-of-the-art performance, especially for encrypted square matrices.
> However, we would like to clarify that the technical goals of our work and [1] are fundamentally different: we focus on designing constant homomorphic multiplications, i.e., multiplying a ciphertext with a plaintext, while [1] aims to handle operations between two ciphertexts.
> We provide a detailed explanation of our technique below, followed by a qualitative and quantitative comparison with [1].
>
> We first explain why our work only involves constant homomorphic multiplication without requiring any ciphertext multiplication, which is also consistent with recent private inference works [2, 3, 4, 5].
> These works use hybrid 2PC protocols, such as secret sharing and homomorphic encryption, and exploit the fact that the model weight $W$ is available by the sever and the input $X$ of each layer is secret-shared between the two parties.
> In details, the server and the client hold the shares $\langle X \rangle_0$ and $\langle X \rangle_1$, respectively, and then the server homomorphically evaluates $W\cdot\mathrm{Enc}(\langle X \rangle_1)$ after receiving $\mathrm{Enc}(\langle X \rangle_1)$ sent by the client.
> Finally, the server sends back $W\cdot\mathrm{Enc}(\langle X \rangle_1) - R$ to the client, where $R$ is a random mask, and set $\langle Y \rangle_0 = R + W\langle X \rangle_0$, while the client decrypts the received ciphertext to obtain $W\langle X \rangle_1 -R$ and sets $\langle Y \rangle_1 = W\langle X \rangle_1 -R$.
> It is not hard to verify $\langle Y \rangle_0 + \langle Y \rangle_1 = W X$.
> Therefore, it is enough to only exploit constant homomorphic multiplications, thereby achieving reasonable performance.
>
> Based on the above explanation, we give a qualitative and quantitative comparison with [1], to further clearly demonstrate the technical differences between them.
> We have also added the comparison with [1] into the related work of the revision.
>
> 1) Qualitative comparison.
> Different from the setting of 2PC private inference, [1] proposes a private *outsourced* inference scheme, which stands for encrypted data and encrypted model.
> To this end, as we illustrate in the above, [1] designs a homomorphic matrix multiplication protocol for multiplying two encrypted matrices.
> It requires to invoke costly homomorphic multiplication and rotation operations, which are about $2 \sim 20\times$ more expensive than the constant homomorphic multiplication operation used in our work (refer to Table 9 of [6]).
> Rather, similar as Cheetah [4], we utilize the property of polynomial multiplication to achieve multiplying an encrypted matrix with a plain matrix without any expensive rotation operation, and hence achieve reasonable performance in the setting of 2PC private inference.
>
> 2) Quantitative comparison.
> We further experimentally compare our solution with [1].
> In the following table, we report the runtime cost (ms) of the two works as the matrix sizes vary, where the results of [1] are obtained from Table 4 in [1].
> From the table, we observe that our solution outperforms the matrix multiplication protocol of [1] by about $5 \sim 14 \times$.
> Notice that it is actually unfair since we execute simpler constant homomorphic multiplication, rather than ciphertext multiplication in [1].
>
> |          |  Dims=4  |  Dims=16  | Dims=64  |
> | :---| :---: | :---: | :---: |
> | [1] | 57.5 | 162.29 | 610.22 |
> | Ours| 10 | 11 | 53 |
> | | | | |

---

> ### Author Response · Authors · 2022-08-02
> **Response to Reviewer QBqi**
>
> ***Q2*. Line 323-324 states, "For linear operations, we can generate in advance Beaver’s triple in the matrix form [21, 36] using our matrix multiplication protocol." Did you mean non-linear operations, unless I am missing something I am unsure why Beaver triples are needed for matrix multiplication?**
>
> *A2*. We apologize for possible misunderstandings due to the brevity of the presentation. Indeed, we can directly use homomorphic encryption to perform matrix multiplication without any additional assistance. However, in the offline/online 2PC setting, this is a typical trick to generate Beaver's triples offline through homomorphic encryption techniques, so that performing matrix multiplication in the online phase only involves lightweight plaintext computations. This can substantially boost computational performance of the online phase. We briefly describe the technical details as follows.
>
> First, we briefly explain the offline/online setting used in recent private inference works like SecureML [7], Minionn [8] and Delphi [9].
> The goal of introducing the offline/online setting is pushing expensive operations into an input-independent offline phase and hence achieving an extremely efficient online performance when the inputs are available.
> To this end, the parties can generate some input-independent correlated randomness using costly cryptographic protocols, like homomorphic encryption.
> Then, these correlated randomness can be consumed in the online phase only with the cost of lightweight non-cryptographic operations.
>
> Next, we illustrate how to generate Beaver's triples in the offline phase.
> The matrix Beaver's triple refers to a triple $(A, B, C)$ satisfying $C = A \cdot B$ where $A$ and $B$ are uniformly sampled.
> To generate this triple, the server and the client randomly and locally sample $A$ and $B$, respectively.
> Then, they jointly invoke our matrix multiplication protocol in Algorithm 1 of the manuscript with inputs $A$ from the server and $B$ from the client, and outputs $\langle C \rangle_0$ to the server and $\langle C \rangle_1$ to the client.
> After that, the server obtains $(A, \langle C \rangle_0)$ while the client obtains $(B, \langle C \rangle_1)$.
>
> Finally, we show how to perform matrix multiplication in the online phase using the generated Beaver's triple.
> Assuming that the parties want to evaluate $Z=X\cdot Y$, where $X$ is owned by the server and $Y$ is owned by the client.
> Given $(A, \langle C \rangle_0)$, the server locally computes $E=X-A$ and then sends $E$ to the client.
> Given $(B, \langle C \rangle_1)$, the client locally computes $F=Y+B$ and then sends $F$ to the server.
> After that, the server obtains the secret share $\langle Z \rangle_0 = F\cdot A - \langle C \rangle_0$ while the client obtains another share $\langle Z \rangle_1 = Y\cdot E - \langle C \rangle_1$.
> It is not hard to verify $\langle Z \rangle_0 + \langle Z \rangle_1 = X \cdot Y$.
>
> ***Q3*. Concurrent work: THE-X: Privacy-Preserving Transformer Inference with Homomorphic Encryption**
>
> *A3*. Thanks to the reviewer for bringing this concurrent work (THE-X) to our attention.
> THE-X and ours share the same goal, which is to enable private inference on Transformers with cryptographically provable security.
> However, we emphasize that this work was published after our submission.
> Below, we illustrate some important differences in terms of protocol design and security.
> We also have added the discussion in the related work of the revision.
>
> 1) Protocol design. Our work aims to design new efficient protocols for the complex operations of Transformers, while orthogonal to ours, THE-X replaces them with HE-friendly operations.
> For example, THE-X replaces GELUs with simpler operations, i.e., ReLUs, and Softmax with the combination of ReLU and polynomials.
>
> 2) Security.
> Our work achieves more rigorous privacy protection than THE-X.
> Specifically, our work uses homomorphic encryption and secret sharing to securely evaluate all layers in ciphertext.
> As a result, the server learns nothing, while the client only obtains the prediction result in the private inference process.
> Such rigorous privacy guarantee is in line with recent state-of-the-art private inference works [3, 4, 5].
> However, in THE-X, the inputs of each non-linear layer are leaked to the client.
> Therefore, our work can be used to enhance the security of THE-X.

---

> > ### Comment · Reviewer_6r55 · 2022-08-06
> > **Concurrent work**
> >
> > Thanks for clarifying the difference. I noticed that the reference [50] was not added to the main content. It would be helpful for other researchers to better understand the work. Overall, I'd like to accept the submission.
> >
> > =====
> >
> > Q3. Concurrent work: THE-X: Privacy-Preserving Transformer Inference with Homomorphic Encryption
> >
> > A3. Thanks to the reviewer for bringing this concurrent work (THE-X) to our attention. THE-X and ours share the same goal, which is to enable private inference on Transformers with cryptographically provable security. However, we emphasize that this work was published after our submission. Below, we illustrate some important differences in terms of protocol design and security. We also have added the discussion in the related work of the revision.
> >
> > Protocol design. Our work aims to design new efficient protocols for the complex operations of Transformers, while orthogonal to ours, THE-X replaces them with HE-friendly operations. For example, THE-X replaces GELUs with simpler operations, i.e., ReLUs, and Softmax with the combination of ReLU and polynomials.
> >
> > Security. Our work achieves more rigorous privacy protection than THE-X. Specifically, our work uses homomorphic encryption and secret sharing to securely evaluate all layers in ciphertext. As a result, the server learns nothing, while the client only obtains the prediction result in the private inference process. Such rigorous privacy guarantee is in line with recent state-of-the-art private inference works [3, 4, 5]. However, in THE-X, the inputs of each non-linear layer are leaked to the client. Therefore, our work can be used to enhance the security of THE-X.

---

> > > ### Author Response · Authors · 2022-08-06
> > > **Response to concurrent work**
> > >
> > > We are sorry for not citing the concurrent work [50] in the main content. We will add it in the revised version. Thanks for your positive review.

---

> ### Author Response · Authors · 2022-08-02
> **Response to Reviewer QBqi**
>
> References:
>
> [1] Secure Outsourced Matrix Computation And Application to Neural Networks, ACM CCS, 2018.
>
> [2] SIMC: ML Inference Secure Against Malicious Clients at Semi-Honest Cost, USENIX Security, 2022.
>
> [3] CrypTFlow2: Practical 2-Party Secure Inference, ACM CCS, 2020.
>
> [4] Cheetah: Lean and Fast Secure Two-Party Deep Neural Network Inference, USENIX Security, 2022.
>
> [5] GAZELLE: A low latency framework for secure neural network inference, USENIX Security, 2018.
>
> [6] nGraph-HE2: A High-Throughput Framework for Neural Network Inference on Encrypted Data, WAHC, 2019.
>
> [7] Secureml: A system for scalable privacy-preserving machine learning, IEEE S\&P, 2017.
>
> [8] Oblivious neural network predictions via minionn transformations, ACM CCS, 2017.
>
> [9] Delphi: A cryptographic inference service for neural networks, USENIX Security, 2020.

---

### Official Review · Reviewer_MFak · 2022-07-11

**Rating:** 7
**Confidence:** 2
**Soundness:** 4 excellent
**Presentation:** 4 excellent
**Contribution:** 4 excellent

**Summary:**

This paper proposes a new secure cryptographic protocol for standard components in Transformers, such as matrix multiplications, Softmax, GeLU, and LayerNorm. The authors' method is the first implementation of secure inference on Transformer for 2PC setup. Furthermore, their new customized homomorphic encryptions allow faster and more efficient matrix multiplications over existing state-of-the-art methods.

**Questions:**

1. In line 198, what are these symbols mean? Typo?
2. Compared to the garbled circuit used in the ReLU protocol from CNN secured inference, is secure GELU protocol par regarding run-time and communication cost to the garbled circuit on ReLU?

**Limitations:**

The authors provide detailed comments on their limitations and possible future directions further to accelerate the private inference on the transformer models.

**Strengths And Weaknesses:**

Strengths
1. Iron is the first private inference framework for the Transformer in secure 2-party computation technique.
2. Iron's custom homomorphic encryptions allow fast linear and non-linear operations by a significant margin compared to the existing state-of-the-art methods such as Cheetah and SIRNN.
3. Provides the protocols for essential components in the Transformer (e.g., Softmax, GeLU) instead of replacing them with easier operations like ReLU.

Weaknesses
1. The authors' submission does not include the official implementation.

---

> ### Author Response · Authors · 2022-08-02
> **Response to Reviewer MFak**
>
> Dear Reviewer,
>
> Thanks for your valuable comments. We address your questions below.
>
> ***Q1*. In line 198, what are these symbols mean? Typo?**
>
> *A1*. We apologize for the possible confusion.
> These symbols are not typos, and $x \nmid y$ means $x$ is not a divisor of $y$. We have clarified it and added the notation in Section 2.2 of the revision.
>
> ***Q2*. Is secure GELU protocol par regarding run-time and communication cost to the garbled circuit on ReLU?**
>
> *A2*. We would like to clarify that the GELU protocol and the garbled circuit (GC) based ReLU protocol serve the secure implementation of different non-linear functions, which derives no natural way to compare the performance of them. We explain it in detail from both theoretical complexity and experimental performance perspectives.
>
> 1) Theoretical complexity. We first recall the formulations of GELU and ReLU, that is, $\mathsf{GELU}(x) = 0.5 x\left(1+ \mathsf{Tanh} \left[\sqrt{2 / \pi}\left(x+0.044715 x^{3}\right)\right]\right)$ and $\mathsf{ReLU}(x) = x \cdot 1$ \{ $x \geq 0$ \}.
> Designing a secure GELU protocol is clearly more complicated than ReLU since it requires to evaluate 1 Tanh, 3 Multiplication, and 3 Scalar Multiplication in ciphertext, while ReLU only involves 1 Comparison and 1 Multiplication. In particular, Tanh is usually evaluated by invoking 1 Comparison, as well as 1 Exponential and 1 Reciprocal, which are costly operations compared to Comparison [1]. Therefore, the inherent differences between GELU and ReLU make it difficult to provide a natural way to compare the performance of them when they are implemented securely.
>
> 2) Experimental performance. Our secure GELU protocol has higher overhead than the GC-based ReLU protocol, which is determined by the high computational complexity of GELU. However, we would like to emphasize that simply implementing GELU in GCs derives undesirable runtime and communication costs compared to our method.
> In the table below, we report the results of our GELU protocol, GC-based ReLU and GC-based GELU, where we use the state-of-the-art GC library, i.e., EMP-Toolkit [2], to implement ReLU and the results of GCs-based GELU are estimated from Table I of SIRNN [1].
> We observe that our GELU protocol is slower than GC-basd ReLU, which is in line with the theoretical complexity.
> However, our GELU protocol achieves at least $5.6 \times$ and  $24 \times$ improvement over the GC-based alternative in terms of runtime and communication  costs, respectively.
> |          |  Time (Sec)  |  Comm. (MB)  |
> | :---| :---: | :---: |
> | GC-ReLU | 0.02 | 1.99 |
> | GC-GELU| 1.68 | 243.46 |
> | Ours-GELU| 0.30 | 10.07 |
> | | | |
>
> ***Q3*. The authors' submission does not include the official implementation.**
>
> *A3*. We have submitted our source code in the supplementary material in the revision, and will open source it in the near future.
>
> References:
>
> [1] SIRNN: A Math Library for Secure RNN Inference, IEEE S\&P, 2021.
>
> [2] EMP-Toolkit: https://github.com/emp-toolkit/emp-sh2pc

---

### Public Comment · ~Wen-jie_Lu1 · 2023-01-09
**Security via adding random mask.**

Algorithm 1 first sample a random matrix $R$ and then parse it as a polynomial.
Then it homomorphically adds it to the resulting ciphertext before sending back to decryption.
However, this pattern is **NOT SECURE** for the coefficient encoding used by Iron since the resulting polynomial
contains more information than the mamat in many coefficients.
So it needs to do it in the inverse direction, ie sample a uniform random polynomial and decode it as one random share.

[update 10-27-2023]
- We have missed the very old paper that have already applied the coefficient encoding over the X^N + 1 ring for matmul
  "Fast Secure Matrix Multiplications over Ring-Based Homomorphic Encryption" by Pradeep et al back to 2018.
   The encoding of Pradeep18 is very similiar to that used by Iron.

   Actually this line of using the coefficient coding for vector inner products start by Yasuda et al. "Secure pattern matching using somewhat homomorphic encryption"

---

> ### Public Comment · Authors · 2023-01-09
> **Response to the security concern**
>
> Dear Dr. Lu,
>
> Thanks for pointing out this issue. Indeed, we worked around this by using the random polynomial sampling and random mask generation methods in your Cheetah implementation. We are sorry that we omitted this detail in our manuscript. We have added this detail in the final camera-ready revision.
>
> Best regards,
>
> Authors

---

### Meta-Review · Area_Chair_1jJc · 2022-08-23

**Recommendation:** Accept
**Confidence:** Certain

**Metareview:**

The paper studies private inference on transformer-based models. It provides methods to securely perform matrix multiplication and certain other non-linear function computation. The evaluations find that their methods are much more efficient than the state of the art.

All of the reviewers are positive about the paper and I recommend acceptance subject to the authors following up on the promised changes (including "We have submitted our source code in the supplementary material in the revision, and will open source it in the near future").

**Award:**

No

---

### Decision · Program_Chairs · 2022-09-14

Accept